# AdaptiVocab: Enhancing LLM Efficiency in Focused Domains through Lightweight Vocabulary Adaptation

**Itay Nakash**[1,*], **Nitay Calderon**[1,*], **Eyal Ben David**[1], **Elad Hoffer**[2], **Roi Reichart**[1]
[1]The Faculty of Data and Decision Sciences, Technion – IIT      [2]Habana Labs

## Abstract

Large Language Models (LLMs) have shown impressive versatility as general-purpose models. However, their broad applicability comes at a high-cost computational overhead, particularly in auto-regressive decoding where each step requires a forward pass. In domain-specific settings, general-purpose capabilities are unnecessary and can be exchanged for efficiency. In this work, we take a novel perspective on domain adaptation–reducing latency and computational costs by adapting the vocabulary to focused domains of interest. We introduce AdaptiVocab, a complete approach for vocabulary adaptation, designed to enhance LLM efficiency in low-resource domains. AdaptiVocab can be applied to any tokenizer and architecture, modifying the vocabulary by replacing tokens with domain-specific $n$-gram-based tokens, thereby reducing the number of tokens required for both input processing and output generation. AdaptiVocab initializes new $n$-token embeddings using an exponentially weighted combination of existing embeddings and employs a lightweight fine-tuning phase that can be efficiently performed on a single GPU. We evaluate two 7B LLMs across three niche domains, assessing efficiency, generation quality, and end-task performance. Our results show that AdaptiVocab reduces token usage by over 25% without compromising performance.[1]

## 1 Introduction

Large Language Models (LLMs) have revolutionized the field of NLP and are now integrated into everyday technologies (Inaba et al., 2023; Chen et al., 2024; Huang et al., 2024). However, their large size and high computational demands lead to significant latency and cost, often hindering their deployment in many real-world scenarios. Practitioners frequently aim to apply LLMs in domain-specific settings (Saad-Falcon et al., 2023; Eschbach-Dymanus et al., 2024; Afzal et al., 2024), where general-purpose capabilities are unnecessary, and efficiency and hardware constraints become critical concerns (Calderon et al., 2023).

The field of domain adaptation addresses the challenge of adapting NLP models to new domains (Blitzer et al., 2007; Ben-David et al., 2022; Lu et al., 2024), typically in low-resource settings with only limited unlabeled domain data (e.g., a few million tokens) (Ramponi & Plank, 2020; Calderon et al., 2022; Marashian et al., 2025). In this work, we take a novel perspective on domain adaptation. Rather than focusing solely on improving end-task performance (Chu et al., 2017; Calderon et al., 2024), we focus on efficiency by reducing computational costs when applying LLMs to new domains. Unlike traditional domain adaptation methods that are model-centric (modifying the architecture or training objective) or data-centric (modifying, augmenting, or refining the data) (Yavuz et al., 2020; Shakeri et al., 2020), we introduce a vocabulary-centric approach that improves efficiency without degrading performance by adapting the vocabulary to fit a target low-resource domain.

LLMs process text by dividing it into predefined tokens composed of subwords and words, a process known as tokenization. The vocabulary, which consists of this set of tokens, is typically selected using widely adopted methods such as Byte Pair Encoding (BPE) (Sennrich

---

[1]Our code and data are available at: `https://github.com/itay-nakash/AdaptiVocab`

| **Mistral-v0.3-7b BPE tokenizer:** 60 tokens | Source: Pérez & Valls (2015) |
|---|---|

Bohr takes advantage of this peculiar fact to devise transformations that connect in a continuous way different stationary states of the same system. In other words, the quantization of adiabatic invariants meets the preconditions for the applicability of Boltzmann's principle.

| **AdaptiVocab (on top of Mistrals tokenizer):** 39 tokens | |
|---|---|

Bohr takes advantage of this peculiar fact to devise transformations that connect in a continuous way different stationary states of the same system. In other words, the quantization of adiabatic invariants meets the preconditions for the applicability of Boltzmann's principle.

Table 1: Tokenization before and after vocabulary adaptation to the *History of Physics* domain.

et al., 2016; Schuster & Nakajima, 2012; Kudo, 2018). Each token is embedded into a high-dimensional vector fed into the transformer layers, thus, an efficient vocabulary selection reduces memory consumption and speeds up processing. In generative auto-regressive models, each generated token requires a forward pass, so optimizing the vocabulary directly reduces decoding steps and improves latency. Since LLM vocabularies are optimized for general-purpose use, domain-specific applications can benefit from replacing redundant tokens with domain-relevant ones, thus enhancing efficiency.

In this paper, we propose AdaptiVocab, the first complete approach for adapting an LLM's vocabulary to a new focused low-resource domain. Figure 1 illustrates the main components of AdaptiVocab. To summarise: (1) We focus on generative decoder-only models and optimize for efficiency; (2) We address monolingual adaptation to low-resource focused domains; (3) We incorporate multi-word tokens into the vocabulary; (4) Our approach is tokenizer-agnostic and works on top of any tokenizer.

AdaptiVocab works as follows: given target domain-specific data, we first compute a frequency-based score for each candidate $n$-token: a combination of $n$ existing tokens that can represent a subword, a single word, or an n-gram. We then modify the vocabulary by replacing low-scoring tokens with high-scoring $n$-tokens. Additionally, we propose an $n$-token tokenization algorithm for encoding and decoding text with the modified vocabulary. Second, we initialize the embeddings of the new $n$-tokens using an exponentially weighted combination of the embeddings of their original constituent tokens, ensuring a smoother integration compared to plain averaging or random initialization. Third, we explore strategies for a lightweight adaptation phase, suggesting fine-tuning only the model's embedding layers and the layers that directly interact with them. AdaptiVocab can be applied to any tokenizer and LLM, requiring minimal computational overhead. The entire adaptation process runs within a few hours on a single RTX A6000 GPU with 48GB of memory.

To evaluate AdaptiVocab, we conducted experiments with two open-source LLMs, Mistral-7B-0.3 (Jiang et al., 2023) and Llama-2 7B (Touvron et al., 2023), across three niche, low-resource domains: Earth Sciences, History of Physics, and Games and Toys. We evaluated AdaptiVocab through LLM-as-a-Judge, human evaluation, and end-task analysis on three new multiple-choice question datasets for our focused domains. Our results show that AdaptiVocab reduces input and output token usage by more than 25% on average without compromising generation quality or end-task performance. We also found that lightweight fine-tuning improves domain-specific performance even for off-the-shelf pre-trained LLMs.

Our contributions: (1) We propose AdaptiVocab, a complete approach for adapting an LLM's vocabulary to a new low-resource domain. Our method encompasses a vocabulary modification algorithm, an embedding initialization technique, a lightweight fine-tuning strategy, and an $n$-token tokenization algorithm; (2) We create three domain-specific question-answering datasets; (3) Our experiments across niche domains demonstrate a 25% efficiency improvement without compromising generation quality or performance. We hope our work will pave the way for practical and efficient vocabulary adaptation of LLMs.

## 2 Related work

**Vocabulary adaptation** Previous research has explored new tokenization strategies to improve language representation and task performance (Uzan et al., 2024; Zouhar et al.,

2023; Schmidt et al., 2024c; Jacobs & Pinter, 2022; Cherf & Pinter, 2024; Schmidt et al., 2024a). However, most of these studies focus on encoder-only models, optimizing for end-task performance rather than generation efficiency (Beinborn & Pinter, 2023; Goldman et al., 2024; Schmidt et al., 2024b). In encoder models, attention layers leveraging contextual information often compensate for weak token representations (Voita et al., 2019; Rogers et al., 2020). In contrast, weak representations in the language model head in generative models degrade generation quality (Press & Wolf, 2017; Yu et al., 2022). Additionally, most existing works focus on general-domain models, which often require complete pretraining (Ma et al., 2020; Boukkouri et al., 2020; Xue et al., 2022; Clark et al., 2022; Pagnoni et al., 2024). Methods for adapting the vocabulary without training from scratch have been explored in the context of cross-lingual adaptation (Minixhofer et al., 2024; Remy et al., 2024; Liang et al., 2023; Downey et al., 2023; Gu et al., 2024; Huang et al., 2024) and coding (Chirkova & Troshin, 2023; Dagan et al., 2024), where the new vocabulary minimally overlaps with the original, necessitating large-scale model training. In contrast, we focus on low-resource monolingual adaptation to a niche domain, where vocabulary overlap is high since both domains share the same language. Sachidananda et al. (2021) examined monolingual adaptation for encoder-only models, selecting new tokens based on their distributional divergence from the general corpus, while we select $n$-tokens based on efficiency. Finally, our work incorporates n-grams into the vocabulary, yielding an additional 10% efficiency improvement (see §5.1).

**Embedding initialization** Prior work has explored transferring embeddings between pre-trained models by preserving token structure and relationships Liu et al. (2021); Sato et al. (2020); Mosin et al. (2023), as well as mapping new vocabulary tokens to existing ones with auxiliary models Dobler & de Melo (2023), such as aligning words with their translations. However, these approaches restrict token selection to words already existing in the vocabularies of other pre-trained models. In contrast, we select new tokens based on a focused target domain, meaning we cannot rely on pre-existing models. Another line of work proposes using hypernetworks to generate embeddings for new tokens (Minixhofer et al., 2024; Feher et al., 2024). However, this approach requires a separate, large-scale training phase and is model-specific since different hypernetworks must be trained for each LLM. Finally, Yamaguchi et al. (2024) compared different initialization techniques for low-resource cross-lingual adaptations of LLMs. They found that initialization with auxiliary models (such as Dobler & de Melo (2023)) performs worse than mean initialization. Our paper introduces exponentially weighting initialization which is suited for generation.

**Generation efficiency** Various methods have been proposed to reduce latency (Treviso et al., 2023; Schwartz et al., 2020; Zhou et al., 2024; Yang et al., 2024; Wan et al., 2024). Most existing research focuses on architectural modifications, such as pruning, and the training of smaller models via knowledge distillation (Calderon et al., 2023). A more closely related approach to ours is multi-token prediction, which generates multiple tokens per step (Gloeckle et al., 2024; Pal et al., 2023; Qi et al., 2020; Cai et al., 2024). Our approach modifies the vocabulary by introducing $n$-tokens and treating each as a single token. This underexplored strategy offers a complementary path for further efficiency gains.

## 3 Method

Our method, AdaptiVocab, is illustrated in Figure 1 and is comprised of four stages: (1) Vocabulary modification: by replacing tokens with domain-specific $n$-tokens that reduces the total tokens count. (2) Tokenization patching algorithm: which ensures our method can be applied to any tokenizer. (3) Embedding initialization: of new $n$-tokens suited for auto-regressive generation. (4) Lightweight adaptation training: of only the embedding matrices and two layers. The final LLM benefits from a reduced token count in both input and output, improving efficiency without degrading performance.

### 3.1 Domain-specific vocabulary modification

Our algorithm for vocabulary modification is presented in Algorithm 1. Given a maximal $n$-token length $n$, a vocabulary $\mathcal{V}_{\text{old}}$, and a domain-specific corpus $\mathcal{D}$, the algorithm replaces $m$ original tokens with $n$-tokens according to a score that quantifies their efficiency (i.e.,

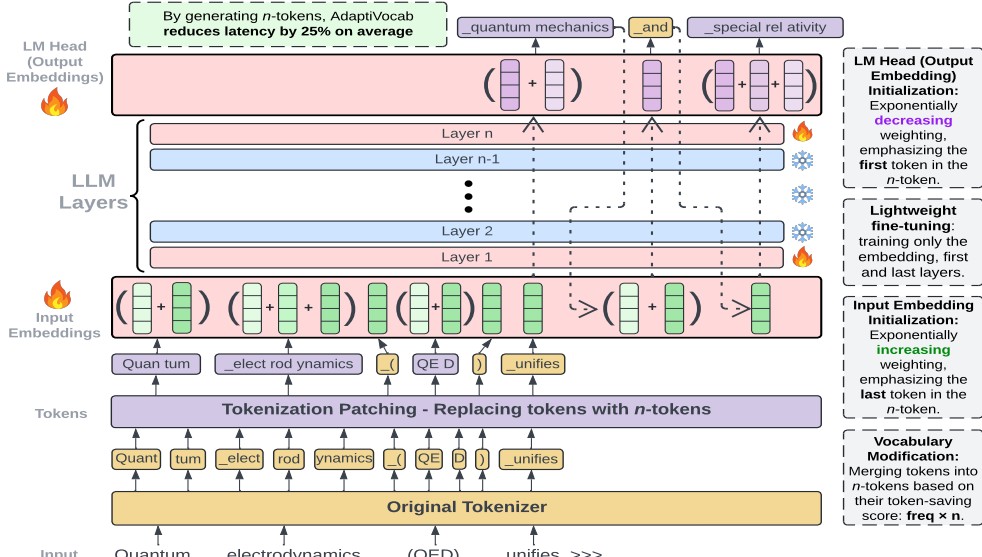

Figure 1: **An illustration of the AdaptiVocab pipeline:** AdaptiVocab adapts the LLM vocabulary to a low-resource domain by replacing general tokens with domain-specific *n*-tokens. To achieve this, it selects *n*-tokens that optimize token savings, initializes the embeddings of new *n*-tokens using exponential weighting, and performs lightweight fine-tuning, yielding 25% efficiency improvement.

their reduction of the total number of units after tokenizing the corpus). Unlike static frequency-based selection of tokens, our algorithm iteratively refines *n*-token scores based on their contribution to efficiency. We initialize the *savings score* of an *n*-token *t*, $S_0[t]$, as the product of its frequency and its length (line 6 in Algorithm 1) and update the score each time we add an *n*-token:

$$S_0[t] = F_{n\text{-tok}}[t] \times \text{len}(t) \qquad S_i[t] = S_0[t] - \sum_{t' \in \mathcal{V}_{\text{new}}^{i-1}} F_{\text{overlaps}}[t][t'] \times \text{len}(t)$$

where $F_{n\text{-tok}}[t]$ holds the number of occurrences of *t* in $\mathcal{D}$, $\text{len}(t)$ is the number of original tokens composing of *t*, $\mathcal{V}_{\text{new}}^{i-1}$ represents the set of $i-1$ new *n*-tokens already added to the vocabulary and $F_{\text{overlaps}}[t][t']$ counts the number of times two *n*-tokens, *t* and *t'*, overlap (we explain this variable below). The $S[\cdot]$ scores capture the token-saving potential of each *n*-token, reflecting the reduction in token count achieved by its inclusion in the vocabulary.[2]

We iteratively replace *m* original tokens with the lowest frequency (line 8 Algorithm 1) with the *m* *n*-tokens that have the highest saving scores (line 9). However, greedily selecting the highest-scoring *n*-tokens may introduce redundancy. To mitigate this, we iteratively update the savings scores of all *overlapping n-tokens* after adding each new one (line 12). Two *n*-tokens are considered overlapping if one contains the other or if the suffix of one matches the prefix of the other. For example, suppose we add the *n*-token special relativity, which consists of special, rel, and ativity (see Figure 1). We then update the frequencies of overlapping *n*-tokens such as special rel (which is contained within special relativity), the special (whose suffix overlaps with the prefix of special relativity), and special relativity theory (which fully contains special relativity). In §5.1 we compare our proposed approach (updating the overlapping token scores) to greedy selection and demonstrate noteworthy savings gains.

When *t* is added, the score of each of its overlapping *n*-token $t' \in F_{\text{overlaps}}[t]$ is reduced by the number of times they co-occur in the corpus (line 12). Specifically, when counting frequencies, we maintain a dictionary $F_{\text{overlaps}}$ that records the number of mutual occurrences

---

[2]The reduction is $F_{n\text{-tok}}[t] \times (\text{len}(t) - 1)$ since we replace $\text{len}(t)$ tokens with a single unit.

| **Algorithm 1** Vocabulary Modification | **Algorithm 2** Tokenization Patching |
|---|---|
| **Require:** tokenizer, corpus $\mathcal{D}$, # new $n$-tokens $m$, max $n$-token length $n$ | **Require:** tokenizer, vocabulary $\mathcal{V}_{\text{new}}$, text |

**Algorithm 1** Vocabulary Modification

1: $\mathcal{V}_{\text{old}}, \mathcal{V}_{\text{new}} = \text{tokenizer.vocab}, \{\}$
2: $\mathcal{D}_{\text{tok}} = [\text{tokenizer}(d) \text{ for } d \in \mathcal{D}]$
3: $\text{n\_tokens} = \text{prepare\_n\_tokens}(\mathcal{D}_{\text{tok}}, n)$
4: $F_{n\text{-tok}} = \text{cnt\_freqs}(\mathcal{D}_{\text{tok}}, \text{n\_tokens})$
5: $F_{\text{overlaps}} = \text{cnt\_overlaps}(\mathcal{D}_{\text{tok}}, \text{n\_tokens})$
6: $S = \{t : F_{n\text{-tok}}[t] \times \text{len}(t) \text{ for } t \in F_{n\text{-tok}}\}$
7: **for** $i = 1 \ldots m$ **:**
8:     $\text{token\_id} = \mathcal{V}_{\text{old}}.\text{pop}()$
9:     $t = \text{argmax}(S)$
10:     $\mathcal{V}_{\text{new}}[t] = \text{token\_id}$
11:     **for** $t' \in F_{\text{overlaps}}[t]$ **:**
12:         $S[t'] -= F_{\text{overlaps}}[t][t'] \times \text{len}(t')$
13: $\mathcal{V}_{\text{new}}.\text{update}(\mathcal{V}_{\text{old}})$
14: **return** $\mathcal{V}_{\text{new}}$

**Algorithm 2** Tokenization Patching

1: **function** DECOMP$(t)$
2:     **if** $t \in \mathcal{V}_{\text{new}}$ **:**
3:         **return** $[t]$
4:     $t_1, t_2 = \text{tokenizer.merge\_table}[t]$
5:     **return** $\text{decomp}(t_1) + \text{decomp}(t_2)$

6: $T_{\text{orig}}, T_{\text{new}} = [], []$
7: **for** $t \in \text{tokenizer}(\text{text})$ **:**
8:     $T_{\text{orig}}.\text{extend}(\text{decomp}(t))$
9: $i, n = 0, max([\text{len}(t) \text{ for } t \in \mathcal{V}_{\text{new}}])$
10: **while** $i < \text{len}(T_{\text{orig}})$ **:**
11:     **for** $k = n \ldots 1$ **:**
12:         **if** $T_{\text{orig}}[i : i + k] \in \mathcal{V}_{\text{new}}$ **:**
13:             $T_{\text{new}}.\text{append}(T_{\text{orig}}[i : i + k])$
14:             $i += k$
15:             **Break**
16: **return** $T_{\text{new}}$

between every pair of overlapping $n$-tokens. This reduction ensures that the savings scores of the remaining candidate $n$-tokens accurately reflect their actual contribution to token saving, after the $i$th $n$-token was added. Note that we assign each new $n$-token a token ID corresponding to a removed token (lines 8 and 10).

## 3.2 Tokenization patching algorithm

We now describe how our tokenization process, which converts a text string into a sequence of tokens, is applied in practice. Standard tokenization algorithms typically operate at the word level, first splitting the text into words and then tokenizing each word separately. However, since our vocabulary includes $n$-tokens, our tokenization must operate at the level of the tokenized text. Accordingly, we introduce a *tokenization patching algorithm* that works on top of any existing tokenizer, involving three stages: (1) tokenizing the text with the original tokenizer, (2) decomposing removed original tokens, and (3) merging tokens into $n$-tokens. The implementation is given in Algorithm 2, and an example is in Figure 1.

We begin by tokenizing the text using the original tokenizer (line 7). We then replace any removed tokens with their corresponding decompositions (line 8). [3] The decomposition process follows the original tokenizer's merging rules and is applied recursively, as defined in the DECOMP function (Algorithm 2, lines 1–5). For example, tokenization is constructed using the rules (*token + iza → tokeniza*) and (*tokeniza + tion → tokenization*). If tokenization and tokeniza are removed, then tokenization is replaced with tokeniza and tion, and subsequently tokeniza is replaced by token and iza, yielding three tokens.

The next part of our tokenization patching algorithm, described in lines 9-15, iteratively replaces spans of original tokens with the longest possible $n$-token. For example, the tokens elect, rod, ynamics from Figure 1 are replaced by the $n$-token electrodynamics. We prioritize merging based on $n$-token length rather than savings score (which involves frequency), as it reduces the total number of tokens for a given text. Additionally, we apply a greedy left-to-right merging strategy because, unlike standard tokenization, which operates at the word level (limited to several characters), our algorithm operates at the tokenized text level. Finding an optimal solution (with dynamic programming that minimizes the total number of units for a given text) would have a computational complexity proportional to the square of the tokenized text length, making it much slower than the original tokenization. Furthermore, the analysis in §5.1 shows that the savings gains of the optimal solution over the greedy left-to-right strategy are negligible. Finally, we map $n$-tokens to their IDs.

---

[3]Note that in practice, token decomposition occurs only rarely, as most removed tokens do not appear even once in the domain-specific corpus. For example, many of them are non-English words.

### 3.3 Exponential embedding initialization

To integrate the newly selected vocabulary, we modify the LLM's embedding layers by replacing removed tokens with new ones. The newly introduced tokens require embeddings in both the input embedding matrix and the language model head (decoder matrix) to ensure they are correctly processed and generated. A straightforward approach to embedding initialization is to assign each new $n$-token the mean of its constituent token embeddings. While simple and previously used (Casanueva et al., 2020; Hofmann et al., 2021; Sachi-dananda et al., 2021; Liu et al., 2023), this *mean initialization* does not effectively address the challenge of auto-regressive generation. It treats the new token as an average of its components but does not account for the structure of the $n$-token.

Since auto-regressive generation produces tokens sequentially from left to right, the input embedding of the last token in an $n$-token should be more dominant, as the next token must coherently continue from it. Conversely, the output embedding (LM head embedding) of the first token in an $n$-token should be more dominant, to increase the likelihood of generating the entire $n$-token and preventing repetitions. Suppose we want to continue the prompt "Quantum electrodynamics unifies" with `quantum mechanics` (see Figure 1). If the output embedding of *the first token* `quantum` is not more dominant than `mechanics`, the model is less likely to generate the entire $n$-token, as `quantum` is the more natural continuation. After generating `quantum mechanics`, the model processes it in the next forward pass using its input embeddings. Ideally, the embedding of *the last token* `mechanics` should be more dominant; otherwise, if both subcomponents contribute equally, the model may incorrectly favor repeating `mechanics` instead of generating a coherent next token.

Accordingly, we introduce the *exponential initialization* that adjusts the influence of constituent token embeddings based on their position (an illustration is given in Figure 1). For input embeddings, the weights increase exponentially ($+$), prioritizing the last token in the $n$-gram to align with how the model processes sequences. For output embeddings, the weights decrease exponentially ($-$), emphasizing the first token to align with generation:

$$\mathbf{e}_{\text{new}} = \sum_{i=1}^{k} w_i \cdot \mathbf{e}_{t_i}, \quad w_i = \frac{e^{\pm 2i}}{\sum_{j=1}^{n} e^{\pm 2j}},$$

### 3.4 Efficient adaptation fine-tuning

Our training process is designed to be lightweight, cost-effective, and accessible, enabling rapid adaptation to domain-specific data. Our embedding initialization method provides a strong starting point for newly introduced $n$-tokens. The fine-tuning process runs on a single RTX A6000 GPU (48GB) for four hours, with an estimated cost of just a few dollars for the entire adaptation process. We fine-tune only a subset of the model's parameters, keeping most transformer layers frozen. Specifically, we update: (1) The input embedding matrix which incorporates the modified vocabulary; (2) The language model head (decoding matrix) to ensure proper $n$-tokens generation; and (3) The first and last transformer layers, which directly interact with the embedding matrices. This strategy allows the model to adapt to vocabulary changes while staying within the 48GB memory limit for 7B-parameter models. We hypothesize that fine-tuning the first and last layers sufficiently aligns the new embeddings with the model's learned parameters while minimizing catastrophic forgetting.

## 4 Experimental set-ups

### 4.1 Models and data

**LLMs** We employed two pre-trained, open-source decoder-only LLMs in our experiments: Mistral-v0.3 (Jiang et al., 2023), which contains 7 billion parameters and a vocabulary of 32,768 tokens, and Llama-v2 (Touvron et al., 2023), which contains 7 billion parameters and a vocabulary of 32,000 tokens. Mistral-v0.3 serves as our primary LLM and is evaluated on three domains, while Llama-v2 complements our results in a single domain. Both LLMs utilize a BPE tokenizer, a common choice among LLMs. Training details, including hyperparameters, are provided in Appendix A. We additionally evaluate input token

savings of five additional LLMs with much larger vocabulary size (up to 262K), including LLaMA 3 (Dubey et al., 2024), Qwen 2.5 (Yang et al., 2025b), Qwen-3 (Yang et al., 2025a), Gemma-3 (Kamath et al., 2025), and DeepSeek-v3 (DeepSeek-AI et al., 2024).

**Baselines**  As discussed in §2, and to the best of our knowledge, no existing method *fully* aligns with our tokenizer-agnostic pipeline, vocabulary modification, embedding initialization, and lightweight training, while also operating under low-resource, hardware-constrained conditions, and this makes direct comparison challenging. Accordingly, we primarily experiment with two baselines: the off-the-shelf (*Vanilla*) LLM and a fine-tuned version trained on domain-specific data (*Vanilla+FT*). We also compare our method (AdaptiVocab+FT) to a variant without fine-tuning (AdaptiVocab). Nevertheless, in §5.1, we compare components of AdaptiVocab to alternative techniques from previous work.

**Datasets**  Our goal is to demonstrate adaptation to focused English-language domains, as opposed to previous works that primarily focus on multilingual or non-natural domains such as programming code. We utilize the M2D2 collection (Reid et al., 2022), which comprises unlabeled datasets from 145 diverse, specialized domains. These datasets vary widely in topic and size, often including domain-specific vocabulary that is underrepresented in general-purpose corpora, making them particularly suitable for evaluating our method. We manually examined dozens of domains from the M2D2 collection, selecting domains that featured proper English text with minimal HTML markup and containing at least 2.5 million tokens. We hence selected three domains: *Earth Science*, *History and Philosophy of Physics* (both have 8.3 million tokens), and *Games & Toys* (2.9 million tokens).

## 4.2 Evaluation

We evaluate our method across four dimensions: generation efficiency, automatic generation quality, human preference, and domain-specific question-answering. The primary target is efficiency, while the remaining dimensions ensure we maintain quality. We briefly describe our evaluation dimensions here, with comprehensive details provided in Appendix C.

**Generation efficiency**  In auto-regressive generative models, each generated token requires a forward pass. Thus, reducing the number of tokens directly improves processing speed and lowers resource consumption. Input length primarily influences memory usage and throughput, with a comparatively smaller effect on overall generation speed (Calderon et al., 2023). We quantify efficiency gains by measuring the percentage reduction in token count. Specifically, we evaluate savings in both *input* (test set) and *output* (see below).

**Automatic generation quality evaluation**  We assess generation quality using 300 test samples per domain (900 total). We truncate test texts by randomly varying lengths, ranging between 15 and 25 tokens, then task each model with generating a 50-token continuation. The completions (the prompt and continuation) are then assessed using an LLM-as-a-Judge, Gemini-1.5-Pro (Reid et al., 2024), that scores the outputs on a scale of 1 to 5 across three key dimensions: logical consistency, coherence, and linguistic acceptability. Appendix D.1 details evaluation prompts and examples.

**Human evaluation**  Human evaluation involved nine annotators (graduate students specializing in NLP) comparing 150 output pairs across domains from Vanilla, Vanilla+FT, and AdaptiVocab+FT. In each comparison, we presented two completions generated from the same prompt, with annotators selecting the better output for each of three key dimensions: logical consistency, coherence, and linguistic acceptability. Further details on the evaluation guidelines are provided in Appendix C.

**Domain-specific question answering**  We created an open-book multiple-choice question dataset for each domain due to the lack of existing datasets. Paragraphs randomly selected from the corpus were used by Gemini-1.5-Pro to generate questions with four answer choices. Generated examples validated by the LLM, and a subset was verified manually and showed high quality. Each dataset contains 100 questions based on paragraphs seen during fine-tuning and 200 from unseen paragraphs (totaling 900 questions). Models received three demonstrations and answered each question, with accuracy as the primary metric. Appendix D.2 provides the generation prompts, and Appendix E.2 presents QA examples.

| Model | Earth Sciences – Mistral-v0.3 | | | | | | Games and Toys – Mistral-v0.3 | | | | | |
|---|---|---|---|---|---|---|---|---|---|---|---|---|
| | Log | Coh | Acpt | Avg | % In | % Out | Log | Coh | Acpt | Avg | % In | % Out |
| Vanilla | 2.78 | 2.58 | 4.08 | 3.15 | 0.0 | 0.0 | 2.09 | 2.46 | 3.76 | 2.77 | 0.0 | 0.0 |
| Vanilla+FT | 2.69 | **2.79** | 4.08 | **3.19** | 0.0 | 0.0 | **2.32** | **2.71** | 3.88 | **2.97** | 0.0 | 0.0 |
| AdaptiVocab | 1.81 | 1.86 | 2.03 | 1.90 | 22.9 | 28.5 | 1.34 | 1.63 | 1.60 | 1.52 | 26.7 | 33.5 |
| AdaptiVocab+FT | **2.95** | 2.35 | **4.19** | 3.16 | 22.9 | 24.9 | 2.15 | 2.01 | **3.92** | 2.69 | 26.7 | 26.5 |

| Model | History of Physics – Mistral-v0.3 | | | | | | History of Physics – Llama-2 | | | | | |
|---|---|---|---|---|---|---|---|---|---|---|---|---|
| | Log | Coh | Acpt | Avg | % In | % Out | Log | Coh | Acpt. | Avg | % In | % Out |
| Vanilla | **2.38** | 2.37 | 3.76 | 2.84 | 0.0 | 0.0 | 2.24 | 2.40 | 4.07 | 2.90 | 0.0 | 0.0 |
| Vanilla+FT | 2.27 | **2.54** | 3.73 | 2.85 | 0.0 | 0.0 | 2.35 | **2.55** | 4.12 | **3.01** | 0.0 | 0.0 |
| AdaptiVocab | 1.47 | 1.43 | 1.79 | 1.56 | 27.9 | 34.5 | 1.54 | 2.13 | 1.65 | 1.78 | 28.6 | 35.8 |
| AdaptiVocab+FT | 2.23 | 2.39 | **4.21** | **2.94** | 27.9 | 27.6 | **2.45** | 2.27 | **4.18** | 2.97 | 28.6 | 27.5 |

Table 2: **Main results – automatic evaluation:** LLM-as-a-judge evaluates three generation-related metrics: Logic (Log), Coherence (Coh), and Linguistic Acceptability (Acpt). Additionally, we report token savings for inputs (% In) and for generated outputs (% Out), which were produced by continuing partial texts from the test set.

## 5 Results

In all our experiments, unless otherwise specified, we use a maximum $n$-token length of three and modify 10,000 tokens. Below, we discuss our main findings:

**AdaptiVocab improves efficiency by 25%** As shown in Table 2, applying AdaptiVocab results in a 22.9–27.9% token reduction when processing input text. Notably, the tokenizer does not see the test set during vocabulary selection. The efficiency gain extends to text generation as well, where token savings are in the 24.9–27.6% range. This reduction directly translates to faster generation, highlighting the practical advantages of vocabulary adaptation. Unlike other approaches that require architectural modifications (e.g., pruning) or extensive training (e.g., knowledge distillation), AdaptiVocab achieves these efficiency improvements with minimal adaptation, making it highly scalable and resource-efficient.

**AdaptiVocab does not compromise generation quality** Despite the token savings, AdaptiVocab maintains competitive text generation quality, as confirmed by both automatic and human evaluations. As shown in Table 2, without fine-tuning, the generation quality is poor due to the introduction of new $n$-token embeddings. However, after lightweight fine-tuning, the LLM effectively adapts, producing outputs comparable to those of vanilla LLMs. Notably, lightweight fine-tuning also benefits vanilla LLMs, which consistently outperform their non-fine-tuned counterparts, reinforcing the importance of domain-specific adaptation. In Figure 2 (full results in Figure 4 in the appendix), we observe that in the majority of pairwise comparisons, human annotators rank AdaptiVocab's outputs as being on par with those of vanilla LLMs. AdaptiVocab performs worst in the Games & Toys domain, which we attribute to its smaller dataset size (3M vs. 8M tokens). Nevertheless, our experiments support the claim that adaptation can be successfully achieved with a moderate dataset of a few million tokens. We also observe that human annotators tend to rate the Vanilla model higher on *acceptability* compared to Vanilla+FT, whereas the trend reverses for *logic* and *coherence*, where fine-tuning provides clear gains. We hypothesize that acceptability reflects general linguistic fluency, which is already well-handled by the pretrained LLM, while logic and coherence depend more on domain-specific reasoning. These latter dimensions might benefit from fine-tuning on the domain data, which explains the observed gap.

**Fine-tuning improves question answering performance** We next assess the impact of vocabulary adaptation on domain-specific knowledge using closed-book multiple-choice question answering. Before that, we first examine whether fine-tuning on domain-specific data can enhance LLM performance on knowledge-based tasks. As shown in Table 3, fine-tuned models (Vanilla+FT and AdaptiVocab+FT) outperform the non-fine-tuned Vanilla model on most seen and unseen QA examples, highlighting the importance of fine-tuning for domain adaptation. Second, regarding vocabulary modification, we observe that AdaptiVocab+FT achieves QA accuracy comparable to Vanilla+FT, demonstrating that vocabulary adaptation does not hinder knowledge retention. This further supports the claim that AdaptiVocab enhances efficiency without compromising performance.

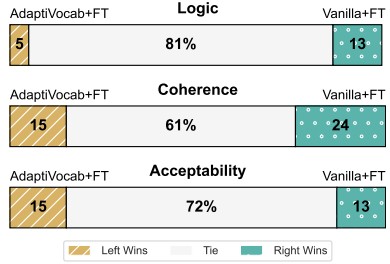

Figure 2: **Human evaluation:** Participants select the better continuations according to three dimensions. Full results in Figure 4.

| Domain | Earth | | Games | | Physics | | |
| Model | Seen | New | Seen | New | Seen | New | Avg |
| --- | --- | --- | --- | --- | --- | --- | --- |
| Vanilla | 0.52 | 0.49 | 0.66 | 0.62 | 0.52 | **0.65** | 0.58 |
| Vanilla+FT | 0.59 | **0.67** | **0.72** | **0.66** | **0.59** | 0.61 | **0.64** |
| AdaptiV | 0.23 | 0.26 | 0.48 | 0.47 | 0.48 | 0.46 | 0.40 |
| AdaptiV+FT | **0.63** | 0.64 | 0.65 | 0.61 | 0.58 | 0.60 | 0.62 |

Table 3: **Multiple-choice QA performance:** Questions were generated from paragraphs in both the training set (i.e., text were Seen during the fine-tuning) and the test set (New unseen texts). The model (Mistral-7b-0.3) selects the correct answer based on the given text.

**Savings in LLMs with larger vocabulary sizes**  While our main experiments focus on two models, LLaMA-2-7B and Mistral-v0.3-7B, we emphasize that our method is tokenizer-agnostic and applicable to any decoder-only LLM architecture. We now provide preliminary evidence of its potential on additional LLMs, particularly those with much larger vocabularies (up to eight times larger). A larger vocabulary could potentially limit our AdaptiVocab's effectiveness, as domain-specific $n$-tokens may already be included, reducing the marginal benefit of vocabulary adaptation. In Table 5, we report input token savings for five additional LLMs after replacing 10K tokens with $n$-tokens. Despite these models having significantly larger vocabularies (128K–262K vs. 32K), we still observe substantial token savings, with at most a 3% reduction compared to smaller-vocabulary LLMs.

## 5.1 Ablation study

We examine the impact of different components, starting with the effects of $n$-token length and the number of modified tokens on token savings, and then evaluating embedding initialization strategies and lightweight fine-tuning techniques.

**Vocabulary modification**  We investigate the effect of the maximum $n$-token length on input token savings and observe that efficiency gains plateau quickly, with no additional savings beyond $n = 4$ (Table 4). Increasing $n$ to 3 provides only a modest improvement of 0.2% (averaged over the three domains), as longer $n$-tokens, occur less frequently, leading to stable overall savings. To further analyze whether token savings stem from merging tokens into single words or forming n-grams, we compare vocabulary modifications constrained to $n \leq 3$. As shown in Table 4, allowing only word-level merges results in 14.5% savings, whereas enabling n-grams achieves 25.6%. Our findings suggest that a vocabulary rich in n-grams is much more efficient than conventional ones. Additionally, in Figure 3 (Appendix), we analyze the impact of the number of replaced tokens and observe that efficiency improves rapidly at first but quickly plateaus. Modifying 10K tokens captures the majority of the savings, which stabilizes around 25%, indicating that most gains can be achieved with a moderate number of vocabulary changes.

We also evaluate two modeling decisions in our approach related to $n$-token selection and merging during tokenization. Recall that our vocabulary modification method involves iteratively updating the savings scores of candidate $n$-tokens that overlap with the most recently added $n$-token. Alternatively, one could adopt a greedy selection strategy that does not update the scores of overlapping tokens. However, as shown in Table 8, our overlap-aware strategy yields up to 1.5% additional token savings, an improvement that is meaningful, especially considering its negligible computational overhead The second modeling decision concerns the use of a left-to-right greedy replacement strategy during the tokenization patching algorithm, where original tokens are replaced with the longest matching $n$-token. While an optimal tokenization can be computed using dynamic programming, this comes at a quadratic computational cost. As shown in Table 9, the optimal strategy yields only a 0.05–0.17% improvement in savings over the greedy approach, confirming that our simpler heuristic is a practical and efficient choice.

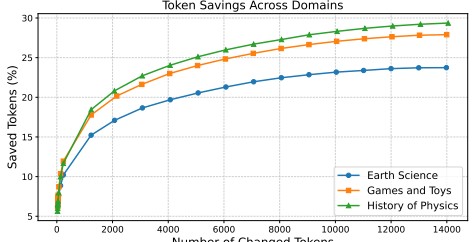

Figure 3: **Impact of the number of modified tokens:** Presenting input savings.

| LLM | Size | Earth | Games | Physics |
|---|---|---|---|---|
| LLaMA 3 | 128k | 21.5 | 24.9 | 26.7 |
| DeepSeek v3 | 128k | 21.0 | 24.9 | 25.8 |
| Qwen 2.5 | 151k | 21.7 | 25.3 | 26.9 |
| Qwen 3 | 151k | 21.7 | 26.9 | 25.3 |
| Gemma-3-12B-IT | 262k | 19.3 | 24.1 | 25.8 |

Table 5: **Additional LLMs:** Input token saving of additional LLMs with larger vocabulary size.

| Domain | $n = 2$ | $\leq 3$ | $\leq 4$ | $\leq 5$ | Words |
|---|---|---|---|---|---|
| Earth | 22.5 | 22.9 | 23.0 | 23.0 | 13.3 |
| Physics | 27.1 | 27.9 | 28.1 | 28.1 | 14.9 |
| Games | 25.7 | 26.7 | 26.8 | 26.8 | 15.4 |
| **Avg** | 25.1 | 25.8 | 26.0 | 26.0 | 14.5 |

Table 4: **Impact of n-token length:** Savings for different lengths and when only words are selected: tokens ($n \leq 3$) can be merged if they form a subword or a word, but not multiple words.

| Embedding Init. | Log | Coh | Acpt | Avg |
|---|---|---|---|---|
| Random +FT | 1.62 | 1.70 | 1.10 | 1.47 |
| Mean +FT | 2.73 | 2.28 | **4.19** | 3.07 |
| Exponential +FT (Ours) | **2.95** | **2.35** | **4.19** | **3.16** |

| Fine-tuning Method | Log | Coh | Acpt | Avg |
|---|---|---|---|---|
| LoRA | 2.41 | 2.38 | 3.43 | 2.74 |
| First Two Layers | 2.91 | 2.29 | 4.09 | 3.10 |
| Last Two Layers | 2.73 | **2.44** | 4.14 | 3.10 |
| First & Last Layers (Ours) | **2.95** | 2.35 | **4.19** | **3.16** |

Table 6: **Ablation study results:** Automatic evaluation results for different embedding initialization techniques (top) and different fine-tuning methods (bottom) for the Earth domain.

**Embeddings initialization** We compare three initialization strategies for new $n$-tokens: random, mean (where each token embedding contributes equally), and exponential. We do not examine other techniques, such as (Dobler & de Melo, 2023), since Yamaguchi et al. (2024) found that for low-resource cross-lingual adaptations of LLMs, initialization with auxiliary models performs worse than mean initialization. In addition, mean initialization is the most common technique (Casanueva et al., 2020; Hofmann et al., 2021; Sachidananda et al., 2021), also used in the closest work to ours (Liu et al., 2023). As shown in the top section of Table 6, exponential initialization achieves the best generation quality, followed by the mean strategy, while random performs the worst.

**Lightweight fine-tuning** We evaluate four fine-tuning techniques, where both the input and output embedding matrices are trained. Many works use adapters for domain-specific fine-tuning (Stickland et al., 2021; Malik et al., 2023), where the most common technique is LoRA (training low-rank adapters in each layer) (Hu et al., 2022; Yamaguchi et al., 2024; Eschbach-Dymanus et al., 2024). We compare LoRA to fine-tuning only the first two layers, only the last layers, and fine-tuning both the first and last layers (ours). We hypothesize that the first and last layers are most crucial, as they interact directly with the modified embedding matrices. As shown in the bottom section of Table 6, fine-tuning both the first and last layers yields the highest generation quality. LoRA performs the worst, followed by fine-tuning only the first layers and then only the last layers. We believe that the last layers are more important because their outputs directly determine which token is generated.

## 6 Conclusions

In this work, we introduced AdaptiVocab, a complete approach for vocabulary adaptation in LLMs, designed to enhance efficiency in domain-specific low-resource settings. AdaptiVocab can be applied on top of any tokenizer, replacing tokens with $n$-gram-based tokens to reduce the number of tokens required for both input and output. Our results show that AdaptiVocab reduces token usage by over 25% while maintaining generation quality and end-task performance. These results are promising, as they demonstrate efficiency gains without compressing the model or altering its architecture – which can also be combined with vocabulary adaptation for further improvements. We hope our work paves the way for future research in this direction.

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

# Appendix

## A  Training details

All experiments were implemented using HuggingFace's Transformers library and PyTorch, executed on a single NVIDIA RTX A6000 GPU with 48GB of memory. The fine-tuning process for each configuration required up to four hours to complete, incurring an estimated cost of just a few dollars. We fine-tuned the LLMs on two domains with 8.3 million tokens each, and one low-resource domain with 2.9 million tokens. We used a sequence length of 768 tokens, a batch size of one, and gradient accumulation over 32 steps. All model layers frozen except for the embedding matrices (input and decoding), and additional two layers, depending on the experiment. The AdamW optimizer was used with a learning rate of $5 \times 10^{-4}$, $\beta_1 = 0.9$, $\beta_2 = 0.95$, and a weight decay of 0.1. The learning rate followed a linear warmup for 500 steps, transitioning to a cosine decay schedule.

## B  Additional Results

At the time of publishing this paper, Liu et al. (2025) demonstrated that training LLMs from scratch using BPE tokenizers that include multi-word tokens can improve efficiency by 27% and lead to better performance on downstream tasks compared to subword-based BPE tokenizers. Unlike our approach, their work involves training both the tokenizer and the LLM from scratch, requiring substantial computational resources. Next, we compare BPE tokenizers with n-grams trained from scratch on focused domain data to our vocabulary modification method. As shown in Table 7, multi-word BPE achieves up to 5% higher input efficiency but modifies over 250% more tokens than our method. In low-resource settings, such extensive changes are difficult to accommodate through cold initialization and lightweight fine-tuning and may even degrade performance. However, in high-resource, general-purpose regimes, as shown by Liu et al. (2025), multi-word BPE improves both efficiency and end-task performance.

| Tokenizer | BPE (with n-grams) | | Ours | |
|---|---|---|---|---|
| | % Save | # Changed Tokens | % Save | # Changed Tokens |
| Earth Science | 27.7% | 26,235 | 22.9% | 10,000 |
| History of physics | 31.2% | 27,368 | 27.9% | 10,000 |
| Games and Toys | 30.9% | 27,440 | 26.7% | 10,000 |

Table 7: **Comparison to BPE:** We compare the input savings of the BPE tokenizer (when n-grams are allowed) trained from scratch on the domain data to our method.

| Domain | With Overlap-Aware Selection | | | Without Overlap Awareness | | |
|---|---|---|---|---|---|---|
| | Earth | Physics | Games | Earth | Physics | Games |
| Mistral-7B | 22.85 | 27.89 | 26.65 | 22.45 | 26.56 | 25.38 |
| Llama-2-7B | 23.41 | 28.59 | 28.02 | 22.95 | 27.18 | 26.50 |

Table 8: **Overlap-aware $n$-token selection:** Comparison of token savings between overlap-aware scoring and naive greedy selection. Updating scores based on token overlaps yields consistently higher savings.

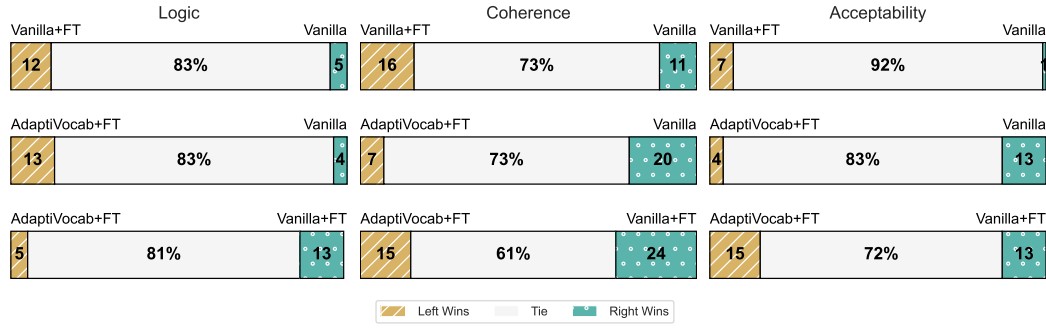

Figure 4: **Human evaluation results:** Human participants were asked to compare two continuations of a prompt and select the better according to three evaluation dimensions.

| Domain | Greedy Tokenization | | | Optimal Tokenization | | |
|---|---|---|---|---|---|---|
| | Earth | Physics | Games | Earth | Physics | Games |
| Mistral-7B | 22.85 | 27.89 | 26.65 | 22.95 | 27.96 | 26.82 |
| Llama-2-7B | 23.41 | 28.59 | 28.02 | 23.47 | 28.64 | 28.13 |

Table 9: **Greedy vs. optimal tokenization:** Comparison of token savings between our greedy left-to-right merging strategy (replacing original tokens with matching $n$-tokens) and a dynamic programming-based optimal replacement. The additional gains from the optimal method are negligible, while its computational cost is quadratic.

## C   Evaluation – additional details

**Automatic generation quality evaluation**   To evaluate generation quality, we generate text continuations based on 300 test samples for each domain (a total of 900 randomly sampled texts). Specifically, we truncate test texts by randomly varying lengths, ranging between 15 and 25 tokens, then task each model with generating a 50-token continuation. The completions (the prompt and continuation) are then assessed using an LLM-as-a-Judge, where Gemini-1.5-Pro Reid et al. (2024) scores the outputs on a scale of 1 to 5 across three key dimensions: logical consistency, coherence, and linguistic acceptability. Appendix D.1 provides details on the evaluation prompts, which include three in-context learning demonstrations. Continuation examples are provided in Appendix E.1.

**Human evaluation**   Alongside automated evaluation, we conducted a human evaluation with nine experts (graduate students specializing in NLP). None of the annotators were among the authors of this paper. Annotators compared 150 pairs of completions across the three domains, assessing outputs from three methods: Vanilla, Vanilla+FT, and AdaptiVocab+FT (ours). In each comparison, we presented two completions generated from the same prompt, with annotators selecting the better output for each of three key dimensions: logical consistency, coherence, and linguistic acceptability. Each pair consisted of paragraphs generated from the same input prefix sampled from the test split. To prevent bias, model identities were anonymized, and both the output order and the order presentation of examples were randomized. Annotators were then instructed to select the better paragraph for each dimension or indicate a tie if no clear preference could be made.

All annotators received standardized guidelines (including a few annotated examples) outlining the evaluation process. The criteria for each aspect were defined as follows:

- **Logical Consistency:** Whether the content presents a sound and non-contradictory line of reasoning.
- **Coherence:** The clarity and continuity of ideas and how well the text flows from one sentence to the next.

- **Linguistic Acceptability:** The degree to which the text adheres to grammar, syntax, and appropriate usage conventions.

**Domain-specific question answering**  This evaluation assesses the ability to retrieve and apply domain-specific knowledge. A key challenge in this setup is the absence of existing end-task evaluation datasets for the three selected domains. To address this, we created an open-book multiple-choice question answering dataset for each domain. We generated these datasets by randomly selecting paragraphs from the domain-specific corpus and using Gemini-1.5-Pro (Reid et al., 2024) to generate a corresponding question with four answer choices, only one of which was correct. Then, we examined all generated examples with the LLM by asking it to check whether the format of the example was valid and to answer the question. We kept only those that were correctly answered by the LLM. A subset of the generated questions and answers were then manually reviewed for quality; we found all the examined examples valid. Each final dataset consists of 100 questions based on paragraphs seen during fine-tuning (models were not fine-tuned on the questions themselves) and an additional 200 questions based on unseen paragraphs (a total of 900 questions, 300 for each domain). For evaluation, we provided the models with three in-context demonstrations, followed by the paragraph and the multiple-choice question, prompting them to select the correct answer. The models' responses were parsed, and accuracy was reported as the primary metric. The prompt template for generating the questions, along with few-shot examples, is provided in Appendix D.2. Examples of generated questions and answers are provided in Appendix E.2.

# D  Prompts

## D.1  LLM-as-a-judge prompts

---

**Box D.1: Logical Consistency Judge Prompt**

Please evaluate the logical consistency of the SCORE below, which continues the provided context.
The text might end abruptly due to length limits; disregard any such truncation in your evaluation.
Logical consistency measures how well the text adheres to logical principles, ensuring that arguments or narratives do not contradict themselves and remain rational throughout.

Rate the logical consistency using a 5-point Likert scale:
1: The text contains multiple contradictions or illogical statements that disrupt understanding.
2: The text occasionally presents contradictions or illogical ideas that hinder comprehension.
3: The text is generally logically consistent, with minor inconsistencies that do not majorly impact the overall logic.
4: The text is mostly logically consistent, with all statements and ideas well-supported and rational.
5: The text is completely logically consistent, with no contradictions, maintaining a rational and coherent argument or narrative from start to finish.

Here are some examples of inputs and ratings:

TEXT: During the Soviet period, the village was a part of the Askeran District of the Nagorno- Karabakh Autonomous Region, where the local economy inexplicably relied on exporting software
technology to ancient civilizations through time travel initiatives.
SCORE: 1

TEXT: Autonomous Region, known for its rich cultural heritage and traditional crafts.  The villagers excelled in carpet weaving and pottery, which were significant contributors to their economy and played a vital role in preserving their culture.
SCORE: 5

---

Now evaluate the following generated sentence:

## Box D.2: Coherence Judge Prompt

Please evaluate the coherence of the generated text below, which continues the provided context.
The text might end abruptly due to length limits; disregard any such truncation in your evaluation. Coherence measures how logically connected and consistent the text is across its entirety.

Rate the coherence using a 5-point Likert scale: 1: The text is very disjointed and lacks any logical connection between ideas.
2: The text has minimal logical connection, with frequent disjointed or unrelated ideas.
3: The text is somewhat coherent, with occasional lapses in logical connection.
4: The text is mostly coherent, with a clear logical flow of ideas throughout.
5: The text is highly coherent, with all ideas logically connected and flowing seamlessly from start to finish.

Here are some examples of inputs and ratings:
TEXT: Max Planck was led by experimental observations to propose a strange formula for the first and time in this theory is the important in the important in time and the universe that in the time of the physical the reality.
SCORE: 1

TEXT: However, whatever the ontology suggested by fundamental physics, one must be able to formulate it in a coherent and precise way. In particular, there is no straightforward translation of this theory into a standard language like, for instance, the language of classical theory..
SCORE: 5

Now evaluate the following generated sentence:

## Box D.3: Linustic Judge Prompt

Please evaluate the linguistic acceptability of the SCORE below, which continues the provided context.
The text might end abruptly due to length limits; disregard any such truncation in your evaluation.
Linguistic acceptability measures how well the text adheres to the norms of grammar, syntax, and usage, ensuring that the language is correctly and appropriately used throughout.

Rate the linguistic acceptability using a 5-point Likert scale:
1: The text frequently uses incorrect grammar, syntax, or inappropriate language that severely disrupts understanding.
2: The text often presents incorrect grammar or syntax, which hinders comprehension and distracts from the content.
3: The text is generally linguistically acceptable, with occasional grammatical or syntactical errors that do not majorly impact readability.
4: The text is mostly linguistically acceptable, with correct use of grammar and syntax, and only minor errors.
5: The text is completely linguistically acceptable, with no grammatical or syntactical errors, using appropriate and precise language from start to finish.

Here are some examples of inputs and ratings:

TEXT: During the Soviet period, the village was a part of the Askeran District of the Nagorno-Karabakh Autonomous Oblast, so roads wasn't paved and nobody didn't pay no mind to what law says about where your sheep.
SCORE: 1

TEXT: The assumption of a unique primary melt led to the expectation that chemical and mineral characterization of primitive glasses associated with a basalt would constrain the residual mantle mineral assemblage responsible for their derivation.
SCORE: 5

Now evaluate the following generated sentence:

## D.2 Domain QA Prompts

### Box D.4: Open Book Question Generation Prompt

Based on this context: (STRAT OF CONTEXT) (Context) (END OF CONTEXT)
Generate a multiple-choice question with four answers.

Format the response as follows:
Question: [question text]
Options:
A) [option 1]
B) [option 2]
C) [option 3]
D) [option 4]
Correct answer: [correct option label (A, B, C, or D)]

### Box D.5: Domain QA answering prompt

Please read the provided context and question carefully, then choose the best answer from the options. Provide the letter of the correct option and the option itself as your answer.

Example 1:
Context: (SOME CONTEXT)
Question: Which of these locations is furthest south?
Options:
A) Dagslett
B) Peja
C) Ivaylovgrad
D) Spikkestad
Answer: C, Ivaylovgrad

Example 2:
Context: (SOME CONTEXT)
Question: Which location was once considered as a potential site for an airport after World War II?
Options:
A) Ivaylovgrad
B) Lyutitsa
C) Gornje Hrasno
D) Ekebergsletta
Answer: D, Ekebergsletta
Now here is your question, answer in the same format presented in this prompt.
Question: ...

# E Examples

## E.1 Generated text examples

---

**Box E.1: Generated text examples - Earth Science**

**Vanilla Mistral:**
*In geology, lamination is a small-scale sequence of fine layers ("laminae"; singular: "lamina") that occurs in sedimentary rocks. Lamination is a type of bedding. Laminae are typically less than 1 mm thick, and are formed by the deposition of sediment in thin layers.*

**Vanilla Mistral Fine-Tuned:**
*In geology, lamination is a small-scale sequence of fine layers ("laminae"; singular: "lamina") that occurs in sedimentary rocks. Lamina can range in thickness from microscopic to macroscopically visible, and can consist of organic fragments or mineral grains.*

**Mistral AdaptiVocab:**

*In geology, lamination is a small-scale sequence of fine layers (ïlaminae; singular: ïlamina) that occurs in sedimented rocks are often interpreted to be to the product of sedimentation, to be the other of material. in the more recent times, to have the formation of 'of the 'lamination.*

*-Layaninctions are of the currents from the ïlauänd ̈cation(G.5: 'washing",*

**Mistral AdaptiVocab Fine-Tuned:**
*In geology, lamination is a small-scale sequence of fine layers ("laminae"; singular: "lamina") that occurs in sedimentary rocks such as shales and limestones. It often describes repetitive patterns in sedimentary strata. Lamination generally begins at the bottom of a stratum, where gravity can create distinct layering. Typical examples of fine laminations are muds, silt, mudstones, and shales.*

---

**Box E.2: Generated text examples - History of Physics**

**Vanilla Mistral:**
*In a sense then, the challenge thrown up by Kirchhoff's observation was to identify and enumerate the list of possible microscopic states of light treated as some kind of substance. Everybody took the Maxwell theory at face value and thought that the task was straightforward to achieve. But Maximalism was about to hit everybody for many reasons: the theory of electromagnetic fields was itself far more complicated than they could have ever imagined.*

**Vanilla Mistral Fine-Tuned:**
*In a sense then, the challenge thrown up by Kirchhoff's observation was to identify and enumerate the list of possible microscopic states of light treated as some kind of substance. Everybody took the Maxwell theory at face value and thought that the state of the electromagnetic field should be characterized by its value at every point of space. But this was obviously too cumbersome and trivially uncountable! It was a well known fact that any.*

**Mistral AdaptiVocab:**
*"In a sense then, the challenge thrown up by Kirchhoff's observation was to identify and enumerate the list of possible microscopic states of light treated as some kind of substance. Everybody took the Maxwell theory, has to a long been to an end uphill battle, The very well theory are not just the result inert particles and in what is a of matter of its is not a and what is made a of something that exists as that is one that the is in the of wells of a theory is a and a is in of light is that of matter.*

**Mistral AdaptiVocab Fine-Tuned:**
*In a sense then, the challenge thrown up by Kirchhoff's observation was to identify and enumerate the list of possible microscopic states of light treated as some kind of substance. Everybody took the Maxwell theory as a basic ingredient so that, in particular, it led to a precise and exact description of the electromagnetic radiation which then was supposed to*

> *be propagating in empty space. However, the basic question was how the basic Maxwell electromagnetic equations were supposed to emerge from the microphysical situation.*

---

**Box E.3: Generated text examples - Games and Toys**

**Vanilla Mistral:**
*"Shadow Knight" does address this inconsistency somewhat, by presenting the "living trump" abilities as somewhat of a "gift" from the gods, rather than a natural ability. The "living trump" abilities are not a natural ability, but a gift from the gods.*

**Mistral AdaptiVocab:**
*"Shadow Knight" does address this inconsistency somewhat, by presenting the "living trum" abilities as somewhat of evil in the strength, and that the "luminous trump" which" is the only way to these skills can be gained. However, none of the descriptions of allude to which one is a character is that has become a clothed in, and which is that is usually is called the evil, with their "luminescence", .*

**Vanilla Mistral Fine-Tuned:**
*"Shadow Knight" does address this inconsistency somewhat, by presenting the "living trump" abilities as somewhat unique to the character. However, it is unclear as to whether this is a distinction with a difference, or simply a way of making the story work.*

**Mistral AdaptiVocab Fine-Tuned:**
*"Shadow Knight" does address this inconsistency somewhat, by presenting the "living trump" abilities as somewhat similar to the "lives" of the classic "Pac-Man".*

---

## E.2   Domain QA examples

**Box E.4: Generated QA example - Earth Sciences**

Context:
Some below this bridge, the Minster receives the waters of the Waag river, whilst a further on, it flows into the southern end of the Sihlsee.
Aghagurty is a townland in County Offaly, Ireland. It is approximately in area. Aghagurty was the ancestral home of Charles Carroll of Carrollton, the only Catholic signatory of the American Declaration of Independence, whose grandfather, Charles Carroll the Settler, is believed to have been born in the locality-his father being known as Daniel Carroll of Aghagurty and Littermurna.
The Winschoterdiep is a canal in the province Groningen of the Netherlands. It leads to the Rensel, which is actually part of this canal. Construction was started in 1618 and finished in 1634. The Winschoterdiep's total length is 35.5 kilometres, and it is approximately 100 metres in width. Sixteen bridges and locks are built across this canal, as well as many other passages. Ships must be less than 16 m in breadth to pass through some of these. It is one of the oldest canals ever built in Groningen still in use.
In the section between Hoogezand and Waterhuizen, there are several shipwharfs. Hoogezand was founded near the canal in 1618.
Mesagne was an important center when Apulia was dominated by the Messapians because it joined Oria to the port of Brindisi. After the Roman conquest, it was also an important city located on the Appian Way. Its name is from these times. In the Middle Ages, it was called "Castrum Medianum", then "Castro Misciano", this is the name used from the 16th century.
When Giovanni Antonio Orsini Del Balzo decided to expand the city's castle, Mesagne evolved, with the construction of a theater, a hospital, and the paving of roads. The city remains important in the economy of the province to this day, with much industry in the area.
Sand is a village in Zala County, Hungary. It is a very small agricultural town, located on gently rolling hills. There is an Evangelical church and a Catholic church near the center of town, and there are memorials to those who served in both world wars nearby. Sand is close to the towns of Csurgo, Iharosbereny, and Lizso, and the largest town nearby is

Nagykanizsa. Adjacent to the church is a wooden bell tower which was built at that same time.

Question:
Which of these locations is described as being situated on gently rolling hills?

Options:
A) Hoogezand, Netherlands.
B) Mesagne, Italy.
C) Aghagurty, Ireland.
D) Sand, Hungary.

---

### Box E.5: Generated QA example - History of Physics

Context:
In spite of the great elegance and simplicity of Newton's treatment of the motion of celestial bodies (with subsequent improvements by Bernoulli and others), the investigation cannot be regarded as complete. The solar system consists of planets, their moons, comets, and asteroids, but the canonical treatment is to consider two bodies only and ignore the rest.
For a long-term prediction of planetary motion, it is necessary to consider additional bodies in the system. The three-body problem was a natural first step and captured the imagination of mathematicians since Newton. Yet other than some restricted situations, no general solution has been obtained. It was considered to be so important that in 1885 King Oscar II of Sweden offered a prize of 2500 crowns for the solution to the three-body problem.
Like others before him, Poincare failed to solve the equations. But unlike others, he solved the problem in a very different sense: he proved that the equations could not be solved. Through attacking the three-body problem, Poincare had laid the foundations for the modern approach to dynamical systems using topology.
Another important application of Newtonian mechanics occurs in the study of the motion of a rigid body. The period formula ( $T = 2\pi \, \text{sqrt}(l/g)$ ) and the solution (a sinusoidal function) that students learn early in school is based on the assumption that the bob is a point oscillating on a plane.
A rigid body problem is a generalization from a point to an extended body rotating in a three- dimensional space. Like the three-body problem, the rigid body problem also attracted the attention of many great mathematicians. Euler studied the case that gravity is indifferent, and Lagrange focused on a symmetric rigid body.
There was no progress for 100 years after Lagrange's work (1788), until Kovalevskaya made a breakthrough. Her result is best described in a letter that she sent to Gosta Mittag-Leffler in 1886: "Dear Sir, I thank you for your invitation for tomorrow, and I shall come with pleasure. It is a question of integrating the following differential equations."
The differential equations in this letter are the Euler equations, which are explained in supplemental material. Kovalevskaya's letter highlights her two achievements. First, she set a new case of the motion of a rigid body and gave a solution in terms of hyperelliptic functions.

Question:
What significant contribution did Sofia Kovalevskaya make to the field of Newtonian mechanics?

Options:
A) She solved the Euler equations for a rigid body where gravity is indifferent.
B) She proved that the three-body problem could not be solved generally.
C) She developed the period formula for a point oscillating on a plane.
D) She provided a solution to a specific case of the rigid body problem using hyperelliptic functions.

**Box E.6: Generated QA example - Games and Toys**

Context:
He launched company Crea-Tech in 1988. Atsuji Yamamoto, Hiroshi Miyaoka's secondary schoolmate, designed for characters; and Satoshi Kadokura contributed music. Tomoki Tauchi, known as the "key man" of the series, directed several "Metal Max" games, also as a programmer of the original "Metal Max." The first "Metal Max" was originally planned to be released before next-generation console Super Famicom's arrival, but it was delayed. It was finally released at the end of the Famicom era, on 24 May 1991, while Super Famicom had been released in November 1990. In the television commercial, the slogan "We've had enough of dragon-slaying" () was used. Compared with "Dragon Quest" and similar games focused on story, "Metal Max" featured an open world similar to Square's "Romancing SaGa." From 1996 to 2005, no new "Metal Max" games were added to the series. After "Metal Max 2" was released, Data East was asked about the third title, but no answer was given by the company. Later, the company went through troubles brought by management issues. Some companies provided offers for developing a Game Boy title. During this period, the Japanese magazine "Super Logo Design" rumored that Crea-Tech would publish "Metal Max 3: Heart of Gold" for the PlayStation. In a 2010 developer meeting, it was said that a PlayStation "Metal Max 3" was conceived but abandoned due to development budget shortages. In 1999, Crea-Tech announced that the sequel would be published for the Dreamcast, tentatively named "Metal Max Overdrive," and planned to be published by ASCII Entertainment, later renamed to "Wild Eyes" and announced for a winter 2000 release. "Wild Eyes" was significantly influenced by the MMORPG "EverQuest" in many aspects, including a full 3D seamless map. This proposal was called "the greatest love story in 'Metal Max' history." However, due to ASCII's poor management, withdrawal from the video game market, and other reasons, the game was canceled. In the late 1990s, Data East ran into financial trouble and sold the games' remake rights to help them survive. Now Production received the rights to remake the SNES titles "Metal Max 2" and "Metal Max Returns" for Game Boy Advance. "Metal Max 2"'s remake version was published on June 20, 2003, and named "Metal Max 2 Kai." "Kai" is literally translated as "modified," referring to the addition of some wanted and rented tanks.'

Question:
What was the primary reason behind the cancellation of "Metal Max Wild Eyes"?

Options:
A) Legal disputes over the remake rights with Now Production.
B) Negative reception to the MMORPG-inspired gameplay.
C) Development budget shortages similar to the PlayStation "Metal Max 3."
D) Poor management and financial troubles at ASCII Entertainment, the planned publisher.

