# OpenReview forum: "AdaptiVocab: Enhancing LLM Efficiency in Focused Domains through Lightweight Vocabulary Adaptation"
_colmweb.org/COLM/2025/Conference — COLM 2025_

### Official Review · Reviewer_j7wR · 2025-04-20

**Rating:** 7
**Confidence:** 4
**Ethics Flag:** 1

**Summary:**

This paper proposes a novel approach to considering n-gram new tokens to accelerate inference in a focused domain, while maintaining comparable performance to off-the-shelf base models. Despite the challenging nature of resource availability for evaluation in a focused domain, this paper conducts an extensive evaluation, using both manual and automatic methods, to demonstrate the effectiveness of the proposed approach. The results clearly show the efficacy of the proposed approach in accelerating inference over 25% without compromising task performance.

**Questions To Authors:**

1. What is the motivation for having a coefficient of 2 in the exponential initialization?

**Reasons To Accept:**

1. **Novel approach in considering n-gram new tokens to accelerate inference in focused domain.**
This paper introduces a novel approach to accelerate inference in focused domains by considering n-gram tokens for partial vocabulary replacement. Unlike methods that introduce new tokens via auxiliary tokenizers, the proposed approach composes n-gram tokens based on existing tokens from the source vocabulary. To initialize new n-gram token weights, it employs exponential initialization. This effectively leverages the characteristics of decoder-based models, as demonstrated by the experimental results, which outperform conventional mean initialization. Furthermore, the overall adaptation method has a marginal impact in task performances according to the results provided.

2. **Extensive evaluation**
This paper rigorously evaluates the proposed approach through extensive manual and automatic methods, including leveraging LLM as a judge. Recognizing the difficulty of experimentation in focused domains due to the limited availability of evaluation datasets, the results nonetheless compellingly demonstrate the superiority of the proposed method over the baseline (i.e. off-the-shelf LLM and its adapted version without vocabulary adaptation).

3. The paper is well-written and easy to read.

**Reasons To Reject:**

1. **Outdated models and their implications for the efficacy over task performance and inference speedups**
The selection of Llama 2 and Mistral models warrants further justification given the availability of more recent architectures such as Llama 3.x, Ministral, Gemma 2/3, and Qwen2.5. Several factors contribute to this concern. Specifically:

    * **Training Data and Robustness**: Newer models (e.g. Llama 3.x and Gemma 2/3) are generally trained on significantly larger datasets. This increased scale is said to lead to reduced robustness in continual pre-training scenarios. This negative impact can also affect the efficacy of the proposed method in recent models in terms of task performance. Does the proposed approach still preserve competitive task performance when applied to such models?

    * **Vocabulary Size and Inference Speed**: The substantial difference in vocabulary size (e.g., 32K for Llama 2 versus 128K for Llama 3) could have implications for the efficacy of inference speedups. I would assume that the larger vocabulary size of more recent models might introduce complexities that affect overall inference efficiency. Clarification on this aspect would be beneficial. (e.g. Does it negatively impact inference efficiency or not?)

2. **Lack of comparison**
The paper provides a comprehensive overview of related studies in vocabulary adaptation. However, it notably omits a comparison to the highly relevant work by Csaki et al. (2023) [https://arxiv.org/abs/2311.05741], which also explores the strategy of replacing less frequent tokens.  How is the effectiveness of the proposed approach compared to this work?

3. **Lack of ablations**
The paper would benefit from an additional ablation analysis to understand the contribution of the vocabulary modification components. Specifically:

    * Scoring Function Effectiveness (Section 3.1): How is the effectiveness of the proposed scoring function in 3.1 over a simpler option (e.g. just consider fertility or frequency)?

    * Overlapping n-tokens vs. Greedy Addition (Lines 150-151): The efficacy of considering overlapping n-tokens instead of using a simpler greedy addition strategy should be empirically demonstrated.

4. **Limited novelty with respect to lightweight fine-tuning**
While the paper highlights a lightweight fine-tuning strategy (training top and bottom layers, embeddings, and LM head) as a contribution, its similarity to methods employed by Remy et al. (2024) and Yamaguchi et al. (2024) warrants further discussion. Both of these works also explore a quite similar training approach, which tunes the initial top and bottom two layers (instead of one for each top and bottom in this work), along with embeddings and the LM head. To accurately contextualize the novelty of the proposed approach, it is essential to:

    * Clearly articulate the specific differences between the presented lightweight fine-tuning strategy and those used by Remy et al. (2024) and Yamaguchi et al. (2024).
    * Include citations to these relevant works within the discussion of the fine-tuning strategy, particularly in L339-348 and any other corresponding parts of the paper.

    Remy et al. (COLM 2024) Trans-Tokenization and Cross-lingual Vocabulary Transfers: Language Adaptation of LLMs for Low-Resource NLP
    Yamaguchi et al. (2024) How Can We Effectively Expand the Vocabulary of LLMs with 0.01GB of Target Language Text?

---

> ### Author Response · Authors · 2025-06-02
> **Official Comment by Authors**
>
> ### References:
>
> [1] Csaki, Zoltan, et al. "Efficiently Adapting Pretrained Language Models to New Languages."
> ‏
>
> [2] Remy, François, et al. "Trans-Tokenization and Cross-lingual Vocabulary Transfers: Language Adaptation of {LLM} s for Low-Resource {NLP." First Conference on Language Modeling. 2024.
> ‏
>
> [3] Yamaguchi, Atsuki, Aline Villavicencio, and Nikolaos Aletras. "Vocabulary expansion for low-resource cross-lingual transfer." arXiv e-prints (2024): arXiv-2406.‏

---

> ### Author Response · Authors · 2025-06-02
> **Official Comment by Authors**
>
> ### Lack of ablations:
> Regarding the scoring function, recall two key aspects that distinguish our work from others:
>
> (1) our focus on efficiency, and (2) the incorporation of n-tokens that can span multiple words (i.e., n-grams).
>
> Under the constraint of a predefined maximum n-token length, optimizing our scoring function would yield better efficiency than any alternative scoring function for input tokenization, assuming no distribution shift between the training and test data.
>
> This is because our scoring function directly measures the number of tokens saved when the training corpus is re-tokenized using our new vocabulary. For output generation, the outcome also depends on how well the LLM learns the data with the new tokenization.
> Fertility-based scores aim to minimize the average number of tokens per word, which can be suboptimal in our setting, as they do not account for n-grams
>
> As shown in Table 2, incorporating n-grams leads to at least a 10% improvement in efficiency.
> Regarding the comparison between greedy n-token selection and applying overlapping score modification, we report below the efficiency improvements.
>
> As can be seen, our overlapping  (OL) modification improves efficiency by up to 1.5%:
>
> | LLM     | Earth Sciences w/ OL | Earth Sciences w/o | History of Physics w/ OL | History of Physics w/o | Games and Toys w/ OL | Games and Toys w/o |
> |:--------|:--------------------:|:------------------:|:------------------:|:----------------:|:----------:|:--------:|
> | Mistral |       22.848        |       22.454       |       27.886       |      26.556      |   26.646   |  25.383  |
> | Llama   |       23.411        |       22.950       |       28.594       |      27.182      |   28.020   |  26.499  |
>
> ---
>
> ### Lightweight fine-tuning:
> The reviewer is right, we missed this point. Indeed, fine-tuning the embeddings, LM head, and adjacent layers was also done in [2, 3].
>
> The main differences between our work and theirs (only on the lightweight fine-tuning aspect) are as follows:
>
> (1) As the reviewer noted, we fine-tune fewer layers.
>
> (2) While their setups are also low-resource, they focus on cross-lingual adaptation, whereas we focus on monolingual adaptation. Validating this approach in a different setting, therefore, provides important empirical evidence. Notably, [2] found this strategy to be the most effective, and so did we.
>
> (3) [1] used this fine-tuning configuration but did not compare it against alternatives; [2] compared it to LoRA. In our work, we also compare first and last layer tuning with tuning the first two and last two layers.
>
> Following your suggestion, we will expand our discussion of these works (already cited in our Related Work section) and better contextualize our method and findings in light of theirs.
>
> ---
>
> ### Coefficient of 2i:
>
> We decided to use ±2i to place more emphasis on the first token (for generation) and the last token (for input embeddings) that make up the n-token.
>
> This choice was based on a manual inspection of outputs from preliminary experiments with AdaptiVocab, conducted without fine-tuning. Our initial aim was to adapt LLMs without fine-tuning; however, as shown in our results, this approach proved ineffective. We found that assigning more weight to the first token (hence using 2i instead of 1i) led to more fluent generations. We will add this clarification to the manuscript. Thank you for raising this question.

---

> ### Author Response · Authors · 2025-06-02
> **Official Comment by Authors**
>
> We thank the reviewer for their thoughtful feedback and for recognizing the novelty and effectiveness of our n-gram-based vocabulary adaptation, as well as the comprehensive evaluation we conducted to support our findings in the focused domains setup.
>
> Below, we address the concerns and questions raised:
>
> ---
>
> ### Newer LLMs:
>
> We thank the reviewer for this valuable point. We agree that evaluating our method on more recent models such as Llama 3 or Qwen 2.5 would further strengthen our findings.
>
> Our method is model-agnostic and applicable to any decoder-only architecture, regardless of the tokenizer. We chose to focus on Llama 2 and Mistral v0.3 because they were among the leading open-source LLMs at the time we began this research.
>
> We did not experiment with LLaMA 3, as the smallest available version is 8B, which exceeds our hardware constraints for fine-tuning.
>
> You raise thoughtful and important arguments regarding newer LLMs.
> Indeed, it has been suggested that the stronger the general-purpose LLM, the more likely it is that continued pretraining may negatively impact general performance.
> However, our work focuses specifically on niche domains, where our primary goal is to improve domain-specific performance rather than maintain general-domain capabilities.
>
> As shown in Table 3, continued pretraining in our setup leads to performance improvements in the target domain. We expect this trend to hold for newer models as well, though verifying this empirically remains important.
> Given the opportunity, we plan to include results for newer models in the final version of the paper.
>
> Regarding the second point, vocabulary size indeed plays a role.
> However, a larger vocabulary does not necessarily imply better coverage of our focused domains.
> For context, we include a table (see below) comparing vocabulary sizes of various LLMs (Mistral v0.3, LLaMA 2, LLaMA 3, Gemma 3, Qwen 2.5, Deepseek v3) alongside the number of their tokens that appear at least five times in each of our target domains.
>
> | LLM             | Vocab size |      Earth Sciences       |      History of Physics       |         Games and Toys        |
> |-----------------|:----------:|:--------------------------:|:--------------------------:|:--------------------:|
> | Mistral         |  32,768    |     17,667 (53.9%)         |       9,834 (30%)          |     8,525 (26%)      |
> | Llama-2-7B      |  32,000    |     16,758 (52.4%)         |      9,351 (29.2%)         |    8,114 (25.4%)     |
> | Llama-3-8B      | 128,000    |     28,893 (22.6%)         |       11,487 (9.0%)        |     9,177 (7.2%)     |
> | Gemma-3-12 B-IT | 262,144    |     35,341 (13.5%)         |       11,054 (4.2%)        |     8,833 (3.4%)     |
> | Qwen 2.5        | 151,643    |     27,506 (18.1%)         |       11,269 (7.4%)        |     9,056 (6.0%)     |
> | Deepseek v3     | 128,000    |     30,684 (24.0%)         |       11,718 (9.2%)        |     9,176 (7.2%)     |
>
> Additionally, given the opportunity, we plan to report the efficiency improvements of our method when applied to newer LLMs, but this will take us more time.
>
> ---
>
> ### Lack of comparison with Csaki et al. 2023:
>
> Thank you for pointing out the work of [1], we missed it and will include it in our Related Work section.
>
> The main difference is that [1] focuses on cross-lingual adaptation in medium-resource settings, whereas our work addresses low-resource monolingual adaptation to focused domains, with an emphasis on improving LLM efficiency. This motivation also drives our incorporation of n-tokens that can span multiple words (i.e., n-grams).
>
> In [1], the authors replace k tokens from the original vocabulary after training a BPE tokenizer with k tokens on the target domain. They evaluate improvements using the fertility metric (i.e., the average number of tokens per word), which, as we understand it, is computed using the Treebank corpus rather than based on token frequency in the target domain. As such, their evaluation focuses more on general language representation quality than on direct efficiency improvements in the target domain. Additionally, their embedding initialization is random.
>
> We provide experiments with and without n-grams, demonstrating that including n-grams improves efficiency by 14.5% to over 25% (see Table 2). We also show that our exponential initialization significantly outperforms random initialization (see Table 4). Nonetheless, we appreciate the relevance of [1] and will discuss it in the revised Related Work section.

---

> ### Comment · Reviewer_j7wR · 2025-06-03
>
> Thanks for the response. It has mostly addressed my concerns. While I acknowledge that it is expensive to run experiments on newer models, it must be worthwhile and make the paper more convincing.

---

> > ### Author Response · Authors · 2025-06-09
> > **Official Comment by Authors**
> >
> > Thank you for the follow-up response. We agree that this could strengthen our work and will make an effort to incorporate such a model into the camera-ready version.
> >
> > As a starting point, we have conducted experiments with several newer LLMs and present the efficiency improvements achieved through input tokenization. These improvements closely mirror those observed in generation, assuming the trends observed in our studied LLMs hold for the newer ones. We are still working on the full training pipeline and evaluation for these models, which we were unable to complete due to resource limitations. However, given the opportunity, we will incorporate the results into the final version.
> >
> > | LLM             | Vocab size | Earth Sciences | Physics Hist | Toys    |
> > |----------------|:----------:|:--------------:|:------------:|:-------:|
> > | Mistral 3      |  32,768    |    22.848      |   27.886     | 26.646  |
> > | Llama 2        |  32,000    |    23.411      |   28.594     | 28.020  |
> > | Llama 3        | 128,000    |    21.506      |   26.681     | 24.872  |
> > | Qwen 2.5       | 151,643    |    21.729      |   26.905     | 25.260  |
> > | Qwen 3         | 151,936    |    21.728      |   25.259     | 26.904  |
> > | Deepseek v3    | 128,000    |    21.047      |   25.781     | 24.851  |
> > | Gemma-3-12 B-IT| 262,144    |    19.300      |   25.781     | 24.050  |
> >
> >
> > As can be seen, the efficiency improvements are roughly similar, though slightly smaller (by 1–2%) for newer LLMs. This may be due to their larger vocabularies. While we replaced 10,000 tokens, this accounts for about 30% of the older LLMs' vocabulary but less than 10% in the newer models.

---

### Official Review · Reviewer_CyPJ · 2025-05-11

**Rating:** 7
**Confidence:** 3
**Ethics Flag:** 1

**Summary:**

This paper proposes to enhance the efficiency of LLMs in focused domains by adapting their vocabulary to lightweight ones. The idea and method are interesting and sound, and the effectiveness is proven by the experiments.

**Reasons To Accept:**

1. The idea of adapting lightweight vocabulary in focused domains for efficiency improvement is interesting.
2. The authors propose a proper suite of methods to address the problem.
3. The experimental results prove the effectiveness of the method.

**Reasons To Reject:**

1. The experiments are performed with relatively old models (Llama2 and Mistral). Why not verify the model on newer models, such as Llama3 and Qwen2.5. It will be more convincing.
2. The experiments show that this approach can lead to 20%-30% acceleration in input and output. For a broader audience, it will be beneficial to illustrate how good this acceleration is by comparing it with the results of other common acceleration practices, such as speculative decoding, etc.

---

> ### Author Response · Authors · 2025-06-02
> **Official Comment by Authors**
>
> We thank the reviewer for their thoughtful and positive feedback. We appreciate your recognition of both the importance and interest of our work, as well for the experimental evidence provided.
>
> Below, we address the concerns and questions raised:
>
> ---
>
> ### Newer LLMs:
> We thank the reviewer for this valuable point. We agree that evaluating our method on more recent models such as Llama 3 or Qwen 2.5 would further strengthen our findings.
>
> Our method is model-agnostic and applicable to any decoder-only architecture, regardless of the tokenizer. We chose to focus on Llama 2 and Mistral v0.3 because they were among the leading open-source LLMs at the time we began this research.
>
> We did not experiment with LLaMA 3, as the smallest available version is 8B, which exceeds our hardware constraints for fine-tuning.
>
> Given the opportunity, we plan to include results for newer models, like Gemma and Qwen 2.5, in the final version of the paper.
>
> ---
>
> ### Comparison to other Acceleration Techniques:
>
> We thank the reviewer for this insightful suggestion and fully agree on the importance of contextualizing our efficiency gains.
>
> As noted in lines 125–127, AdaptiVocab is compatible with decoding-level acceleration techniques, such as speculative decoding and multi-token prediction, and can therefore provide additional savings on top of these methods.
>
> Regarding comparisons to speculative decoding, according to the empirical survey in [1], speculative methods improve latency by a factor of 1.37 to 1.74 (for temperature T=1, depending on the method), corresponding to a reduction of approximately 27% to 42.5%.
> Notably, [1] highlights that the gains vary substantially across tasks and draft models.
> For instance, summarization tasks-where large portions of text can be copied from the source, benefit more from speculative sampling than QA tasks. In our focused-domain setup, we hypothesize that the gap between the LLM and the draft model could be even larger than in the general-domain settings evaluated in [1], leading to more rejections of draft tokens and thus smaller gains from speculative decoding.
>
> Analyzing the interaction between speculative decoding and vocabulary adaptation in focused domains is a promising direction for future work, and we will add a discussion of this topic to the paper.
>
> ---
>
> ### References:
>
> [1] Xia, Heming, et al. "Unlocking efficiency in large language model inference: A comprehensive survey of speculative decoding." arXiv preprint arXiv:2401.07851 (2024).‏

---

### Official Review · Reviewer_6fu8 · 2025-05-13

**Rating:** 6
**Confidence:** 4
**Ethics Flag:** 1

**Summary:**

This paper presented an end-to-end approach for vocabulary adaptation for language models, called AdaptiVocab, that is tailored to make encoding/decoding process efficient in low-resource domain settings. The authors have proposed a new approach to create a targeted-domain vocabulary by replacing tokens with the domain-specific n-gram-based tokens. The proposed approach successfully reducing the number of tokens in both encoded and decoded sequences. They also explored the initialization of the new embeddings and lightweight finetuning on a single GPU. Experiments with two different LLMs show that the proposed approach successfully reduce the token usage by 25% without losing performance.

The idea of n-gram-based vocabulary adaption is straightforward. The proposed exponential initialization method appears both interesting and effective, as seen by the results in Table 4. However, the experiments are one-sided and lacks comparison against the previous work, resulting in less technically sound. It'd be great if the authors conducted the established downstream tasks such as (multilingual) QA tasks (Bandarkar et al.), to see how the proposed approach is robust across languages.

- Bandarkar et al. "The belebele benchmark: a parallel reading comprehension dataset in 122 language variants." In Proc of ACL 2024.

For better readability
- l.115 I'd recommend minimizing unnecessary ambiguity when possible. For example:
 "Finally, Yamaguchi et al. (2024) compared different initialization techniques for low-resource cross-lingual adaptations of LLMs. They found that initialization with auxiliary models (such as Dobler & de Melo (2023)) performs worse than mean initialization."  -> "Finally, Yamaguchi et al. (2024) compared different initialization techniques for low-resource cross-lingual adaptations of LLMs and found that initialization with auxiliary models (Dobler & de Melo, 2023) performs worse than mean initialization."
- l.136 Given a maximal n-token length n, .... -> Given a maximum n-token length n, ....
- l. 266 900 total -> 900 in total
- l.325 n-gram -> $n$-gram
- l.337 "also used in the closest work to ours (Liu et al., 2023)" -> can you differentiate the work (Liu et al., 2023) from yours precisely?


Missing related work:
- Han et al., "Adapters for Altering LLM Vocabularies: What Languages Benefit the Most?", In Proc of ICLR 2025.

**Questions To Authors:**

- How did you determine the exponential initialization equation, specifically e^(±2i)? What about e^(±i) or so?

- Have you ever conducted any preliminary experiments by using some established downstream tasks in less domain specific settings? If yes, it'd be worthwhile reporting the results. The current reported tables are a little confusing, hard to say how effective the proposed approach is.

- The authors could also extend the experiments to multilingual settings. A language without whitespace, such as Chinese corpora, would be good to explore as additional results.

**Reasons To Accept:**

The idea of n-gram-based vocabulary adaption is straightforward. Some experimental results look interesting, e.g., the proposed exponential initialization method appears both interesting and effective.

**Reasons To Reject:**

The paper is sometimes hard to follow, due to the format issues. The font style in the manuscript looks different from the COLM paper template, and I'd appreciate if you could double-check the webpage (https://colmweb.org/cfp.html), saying "To prepare your submission to COLM 2025, please use the LaTeX style files provided at: https://github.com/COLM-org/Template/archive/refs/tags/2025.zip". Additionally, there is room to improve its readability (Please see the summary for the details.)

Also, please see the summary.

---

> ### Author Response · Authors · 2025-06-02
> **Official Comment by Authors**
>
> ### References:
>
> [1] Han, HyoJung, et al. "Adapters for Altering LLM Vocabularies: What Languages Benefit the Most?." The Thirteenth International Conference on Learning Representations.
> ‏
>
> [2] Liu, Siyang, et al. "Task-adaptive tokenization: Enhancing long-form text generation efficacy in mental health and beyond." arXiv preprint arXiv:2310.05317 (2023).
> ‏
>
> [3] Stap, David, et al. "The Fine-Tuning Paradox: Boosting Translation Quality Without Sacrificing LLM Abilities." Proceedings of the 62nd Annual Meeting of the Association for Computational Linguistics (Volume 1: Long Papers). 2024.
> ‏
>
> [4] Zhang, Zheng, et al. "Balancing specialized and general skills in llms: The impact of modern tuning and data strategy." arXiv preprint arXiv:2310.04945 (2023).
> ‏
>
> [5] Barnett, Scott, et al. "Fine-tuning or fine-failing? debunking performance myths in large language models." arXiv preprint arXiv:2406.11201 (2024).‏

---

> > ### Comment · Reviewer_6fu8 · 2025-06-08
> >
> > Thank you for the thorough response. It addressed my concerns and I've updated my score accordingly.

---

> > > ### Author Response · Authors · 2025-06-09
> > > **Official Comment by Authors**
> > >
> > > Thank you for the follow-up and for updating your score. We appreciate your time and feedback, and we believe your suggestions and clarifications will help improve our camera-ready version.

---

> ### Author Response · Authors · 2025-06-02
> **Official Comment by Authors**
>
> First, we thank the reviewer for their valuable feedback and helpful suggestions to improve the readability of our paper. We will ensure these suggestions are incorporated. Regarding the missing related work [1], you are right, we overlooked this relevant paper. It focuses on cross-lingual adaptation in high-resource settings, whereas our work targets low-resource monolingual adaptation to focused domains. We will include it in the Related Work section.
>
> Below, we address the concerns and questions that have been raised:
>
> ---
>
> ### Format issues:
> Unfortunately, we copy-pasted the package import commands from another paper we wrote using the ACL template. One of these commands was \usepackage{times}, which unintentionally changed the font of the entire paper, and we didn’t notice the difference. We only became aware of it after submission and have already fixed it. For clarity, this change has no impact on the paper's length; the content and layout remain the same, with only the font style affected.
>
> ---
>
> ### Differences from Liu et al., 2023 [2]:
>
> This work is the most closely related to ours, as it also focuses on adaptation to a single, focused domain (mental health), rather than cross-lingual or code adaptation, and it considers n-grams.
>
> The key differences are:
>
> (1) They experiment with small pre-LLM models like BART and GPT-2, enabling full fine-tuning.
>
> (2) Their primary focus is on improving task performance, rather than efficiency.
>
> (3) They extend the vocabulary by training a unigram tokenizer on the target domain and adding tokens to the original vocabulary, without replacing existing tokens or aiming to improve efficiency, unlike our objective. Moreover, this approach works for unigram tokenizers, whereas our approach can be applied on top of any tokenizer.
>
> (4) They only use mean initialization.
>
> ---
>
> ### Exponential equation:
>
> We decided to use ±2i to place more emphasis on the first token (for generation) and the last token (for input embeddings) that make up the n-token. This choice was based on a manual inspection of outputs from preliminary experiments with AdaptiVocab, conducted without fine-tuning. Our initial aim was to adapt LLMs without fine-tuning; however, as shown in our results, this approach proved ineffective. We found that assigning more weight to the first token (hence using 2i instead of 1i) led to more fluent generations.
>
> We will add this clarification to the manuscript. Thank you for raising this question.
>
> ---
>
> ### General domain and monolingual settings:
> We would like to reiterate the primary setup of our paper: low-resource adaptation to a focused domain.
>
> Specifically:
>
> 1. We address monolingual adaptation, rather than cross-lingual settings (including coding), which are the primary focus of most related work.
> 2. We target adaptation to a narrow, focused domain, based on a specific corpus that includes domain-specific jargon.
> 3. We operate under limited resources, with only a few million unlabeled tokens (~3–8M).
>
> We intentionally chose to focus on this setup because we believe it reflects practical industrial scenarios, where practitioners seek to adapt models to their own domain-specific data.
>
> This setup is notably different from most prior work on vocabulary adaptation (which focuses on cross-lingual and coding) and remains relatively underexplored.
>
> Accordingly, we did not aim to improve performance on downstream, domain-agnostic tasks, to keep the paper’s focus clear.
> We did run preliminary experiments on general-domain tasks and found that both Vanilla+FT and AdaptiVocab+FT performed worse than the original vanilla model. This was expected, as both LLMs had been fine-tuned on domain-specific data.
>
> While we can add this clarification to the paper, we are cautious about shifting the main focus of the paper. Moreover, performance degradation on general-domain tasks after domain-specific fine-tuning is a well-known open problem in NLP (e.g., [3, 4, 5]), and our paper does not aim or claim to address it.
>
> Our method is not designed to perform as well as specialized methods for cross-lingual adaptation.
> We select new n-tokens by merging a few (up to 3-4) existing tokens. In cross-lingual settings, especially when the original tokenizer has limited coverage of the target language, new “optimal” tokens may correspond to long sequences (e.g., a new token might consist of only characters from the new language), making our selection algorithm less suitable.
>
> In such cases, training a tokenizer from scratch (as done in most such works) on the target language or borrowing a vocabulary from another LLM is likely a better approach.
> Once again, cross-lingual adaptation is not the focus of our work, and we do not aim to address it in this paper.

---

### Official Review · Reviewer_Rk2K · 2025-05-13

**Rating:** 6
**Confidence:** 4
**Ethics Flag:** 1

**Summary:**

The authors introduce an approach to adapt the vocabulary of LLMs to low-resource domains by replacing tokens with domain-specific n-gram based tokens. The new embeddings are initialized using an exponentially weighted combination of existing embeddings. Only a couple of layers touching the embeddings are fine-tuned.
The results show that token usage is reduced by 25% with small to no degradation in performance.

**Questions To Authors:**

- Have you tried other weighted initialization methods for the embeddings? Random and Mean are very simple baselines.
- Since the method is suboptimal and relies on greedy heuristics, it would be nice to compare it to the optimal (but inefficient) topline.
- It would be nice to quantify the effect of having test tokens that are seen/unseen during training. Do you have a breakdown of the results in this dimension?
- Human evaluations clearly favor Vanilla+FT for Logic and Coherence, but the scores are close for Acceptability. Could you comment on this?

**Reasons To Accept:**

The paper is clearly written. Figures, algorithms and examples add to the clarity.
The approach is well motivated and reasonable.
The reduction in token usage is significant.

**Reasons To Reject:**

The experimental results lack a baseline vocabulary adaptation approach. The comparisons are made to Vanilla and Vanilla+FT models.

The authors claim that the approach is "end-to-end", which (to me) would require the vocabulary modification and the other steps to use the "end-to-end" loss. This is clearly not the case.

---

> ### Author Response · Authors · 2025-06-02
> **Official Comment by Authors**
>
> ### References:
>
> [1] Mosin, Vladislav D., Igor Samenko, Borislav Kozlovskii, Alexey Tikhonov, and Ivan P. Yamshchikov. "Fine-tuning transformers: Vocabulary transfer." Artificial Intelligence, vol. 317, p. 103860, 2023.
>
>
> [2] Ozeren, Enes, Yihong Liu, and Hinrich Schütze. "HYPEROFA: Expanding LLM Vocabulary to New Languages via Hypernetwork-Based Embedding Initialization." arXiv preprint arXiv:2504.21018, 2025.
>
>
> [3] Han, HyoJung, et al. "Adapters for Altering LLM Vocabularies: What Languages Benefit the Most?." The Thirteenth International Conference on Learning Representations.
> ‏
>
> [4] Minixhofer, Benjamin, Edoardo Maria Ponti, and Ivan Vulić. "Zero-Shot Tokenizer Transfer." Advances in Neural Information Processing Systems 38 (NeurIPS 2024), Vancouver, Canada, Dec. 10–15, 2024.
>
>
> [5] Dobler, Konstantin, and Gerard De Melo. "FOCUS: Effective Embedding Initialization for Monolingual Specialization of Multilingual Models." Proceedings of the 2023 Conference on Empirical Methods in Natural Language Processing. 2023.
> ‏
>
> [6] Liu, Siyang, et al. "Task-adaptive tokenization: Enhancing long-form text generation efficacy in mental health and beyond." arXiv preprint arXiv:2310.05317 (2023).
> ‏
>
> [7] Yamaguchi, Atsuki, Aline Villavicencio, and Nikolaos Aletras. "Vocabulary expansion for low-resource cross-lingual transfer." arXiv e-prints (2024): arXiv-2406.‏

---

> ### Author Response · Authors · 2025-06-02
> **Official Comment by Authors**
>
> ---
> ### Q2: Greedy vs. Optimal
>
> There are two components in our method where we employed suboptimal, greedy-like strategies.
> First, in the selection of n-tokens: selecting an optimal subset of n-tokens that maximizes the total saving score is an NP-hard problem. Given the tens of thousands of candidate n-tokens, exhaustive search is computationally infeasible. Our approach, while greedy in nature, is not naive. We adjust the score of overlapping tokens during the selection process (see lines 149-162). Specifically, after adding each n-token, we update the saving scores of remaining candidates to reflect the current tokenization state.
> Below, we compare the effect of our approach (updating overlapping (OL) token scores) to greedy selection. As can be seen, this yields up to an additional 1.5% improvement.
>
> | LLM     | Earth Sciences w/ OL | Earth Sciences w/o | History of Physics w/ OL | History of Physics w/o | Games and Toys w/ OL | Games and Toys w/o |
> |:--------|:--------------------:|:------------------:|:------------------:|:----------------:|:----------:|:--------:|
> | Mistral |       22.848        |       22.454       |       27.886       |      26.556      |   26.646   |  25.383  |
> | Llama   |       23.411        |       22.950       |       28.594       |      27.182      |   28.020   |  26.499  |
>
>
> Second, in the tokenization process (i.e., converting strings into tokens), we currently apply a greedy approach by always selecting the longest matching n-token at each step.
> While an optimal tokenization can be computed using dynamic programming, it has quadratic complexity in the number of tokens, resulting in significant “real-time” latency (see lines 184-188).
>
> Below, we compare our greedy tokenization approach to the optimal solution obtained via dynamic programming. As can be seen, the optimal solution achieves only a 0.05%-0.17% improvement in efficiency over our method, but it introduces significant latency in the tokenization process. For this reason, we chose not to use it.
>
> | LLM     | Earth Sciences greedy | Physical Science optimal | History of Physics greedy | History of Physics optimal | Games and Toys greedy | Games and Toys optimal |
> |:-------:|:---------------------:|:------------------------:|:--------------------------:|:---------------------------:|:---------------------:|:----------------------:|
> | Mistral |        22.848         |          22.954          |          27.886            |           27.962            |        26.646         |         26.817         |
> | Llama   |        23.411         |          23.472          |          28.594            |           28.641            |        28.020         |         28.130         |
>
>
>
> ---
>
> ### Q3: Seen vs. Unseen Tokens at Test Time
>
> We would like to highlight an important property of our method: all new n-tokens are selected solely based on their frequency (saving score) in the training corpus.
> By construction, this ensures that every modified or replaced token appears in the training data and there are no unseen n-tokens.
>
> We can provide an analysis comparing the frequency of each n-token in the training data versus its frequency in the generated text. Please let us know if this is what you had in mind.
>
> Alternatively, if you were referring to the distinction between “seen” and “unseen” in the QA evaluation, this refers to whether the questions were generated from paragraphs that were included in the training set (seen) or from held-out test data (unseen).
>
> ---
>
> ### Q4: Human Evaluation – Acceptability vs. Logic/Coherence
>
> Thank you for raising this interesting point. We observed the difference, but did not have the space to address it in the main paper.
>
> We hypothesize that logic and coherence are more closely tied to domain-specific knowledge, which benefits from fine-tuning on the domain corpus; hence, the improvements seen with the fine-tuned models. In contrast, acceptability reflects general linguistic fluency, which the original LLM already handles well, and therefore is superior to the vanilla-finetuned model (which is modified).

---

> ### Author Response · Authors · 2025-06-02
> **Official Comment by Authors**
>
> We thank the reviewer for their thoughtful review and for recognizing the paper's strengths.
>
> ---
>
> ### Clarification on “End-to-End”
>
> We appreciate this feedback and agree that our use of the term “end-to-end” may be misleading.
>
> Our intention was to convey that our pipeline encompasses all necessary steps for vocabulary adaptation, rather than focusing on a single component (e.g., reusing embeddings from another LLM vocabulary).
> We used the term “end-to-end” in a more general sense, meaning that it covers the entire process from start to finish.
>
> However, we acknowledge that in ML, “end-to-end” implies joint optimization of the entire system, which is not the case in our approach. To avoid confusion, we will rephrase this description in the manuscript. Thank you for pointing this out.
>
> ---
>
> ### Lack of Baselines + Q1: Embedding Initialization
>
> We would like to reiterate the primary setup of our paper: low-resource adaptation to a focused domain. Specifically:
> 1. We address monolingual adaptation, rather than cross-lingual settings (including coding), which are the primary focus of most related work.
> 2. We explore adaptation to a narrow domain using a targeted corpus containing domain-specific jargon.
> 3. We operate under limited resources, with only a few million unlabeled tokens (~3-8M).
> 4. We focus on efficiency.
> 5. We want a method that can be applied on top of any LLM regardless of the architecture or tokenizer.
>
> Existing methods that perform vocabulary adaptation are typically designed for cross-lingual or multilingual settings, where the vocabulary shift is more drastic and requires extensive model retraining with large-scale data. Moreover, most related work focuses on a specific component of the adaptation process, mainly on embedding initialization, and does not incorporate n-grams (tokens that span multiple words). Yet, we can compare different components of our method.
>
> **Vocabulary selection:**
>
> Our focus is on efficiency. Since the original and target domains have substantial overlap (unlike in cross-lingual or code adaptation), we aimed to further improve efficiency by incorporating n-tokens that span multiple words (i.e., n-grams) into the vocabulary. This design choice led to more than a 10% gain in efficiency compared to vocabulary adaptation using only unigrams (see Table 2).
>
>
> The work most similar to ours in terms of monolingual adaptation to a focused domain (mental health) and the use of n-grams is [6]. However, there are several key differences.
> That work proposes a method based on unigram tokenizers, whereas our approach is compatible with any tokenizer, and in particular with BPE tokenizers that are most popular in modern LLMs.
> [6] train a unigram tokenizer from scratch on the target domain and extend the original vocabulary, while we selectively replace tokens. They use mean initialization and full fine-tuning on GPT-2 and BART, which are considerably smaller than LLMs. Although we do not perform a direct comparison due to these differences, we do evaluate variants of components from their setup. Specifically, in Appendix B, we train BPE tokenizers *with n-grams* from scratch on the target domains. These experiments yield efficiency improvements of 27.7–31.2%, compared to our improvements of 22.9–27.9%. However, training BPE from scratch replaces roughly 27K tokens (out of a 32K vocabulary), whereas we replace only 10K. Such drastic changes, combined with limited training data and lightweight fine-tuning, result in LLMs that generate gibberish, so we excluded those results from the main paper. Finally, we also compare their mean initialization approach to our exponential initialization, as discussed below.
>
>
> **Embedding initialization (+ your first question):**
>
> These constraints above have two important implications:
> 1. We cannot rely on embeddings from external models for initialization. Since our adaptation is monolingual and focused on a niche domain, there is no external tokenizer or model trained on the same domain, unlike cross-lingual or code adaptation scenarios.
>
> 2. We are unable to perform extensive training due to the limited size of our dataset.
>
> As a result, the number of applicable initialization baselines is limited.
> Many related works (e.g. [1],[2],[3]) rely on large-scale training (using billions of tokens) or use embedding transfer from external models trained on similar data ([4],[5]), neither of which is relevant to our setting.
>
> We chose to compare against mean initialization for two main reasons (see Lines 331–338):
> It is used in the closest prior work to ours (e.g., [6]). Prior work [7] on *low-resource* cross-lingual adaptation shows that mean initialization, while trivial, outperforms initialization using auxiliary model embeddings (e.g., FOCUS of [5]). This makes mean initialization a fair baseline for our setting.
>
> **Lightweight fine-tuning:**
>
> We compare our method to LoRA and partial layer fine-tuning strategies in Table 4.

---

> > ### Comment · Reviewer_Rk2K · 2025-06-08
> >
> > Thank you for your response.
> >
> > Clarification on “End-to-End”: I am happy to hear that you agree.
> >
> > Lack of Baselines + Q1: I still think the lack of an alternative baseline is a shortcoming. Q1 was more of a curiosity.

---

> ### Comment · Reviewer_Rk2K · 2025-06-08
>
> Thank you for your response to my comments/questions.
>
> Q2: I think it is informative that "the optimal solution achieves only a 0.05%-0.17% improvement in efficiency over our method, but it introduces significant latency in the tokenization process." This might be worth adding to the paper as other readers may also ask the same question.
>
> Q3: I was referring to those words in the test set that were seen versus unseen during training. There may or may not be any difference in performance but it would be nice to know.
>
> Q4: Your hypothesis is reasonable.

---

> > ### Author Response · Authors · 2025-06-09
> > **Official Comment by Authors**
> >
> > Thank you for the helpful response. We will incorporate your suggestions and clarifications into the camera-ready version, including changing the end-to-end terminology, the comparison between the optimal and greedy tokenization inference methods, and the discussion of the human evaluation results.
> >
> > We would like to include the additional analysis you suggested (Q3), however, we would appreciate further clarification regarding the exact comparison you would like us to conduct.
> >
> > Thank you again for your feedback and engagement throughout the discussion period.

---

### Decision · Program_Chairs · 2025-07-08

**Decision:**

Accept

**Comment:**

This paper introduces the idea of replacing multiple tokens with a single token to improve efficiency in specialised domains. The benefit comes from the fact that otherwise some of the vocabulary specific to that domain is broken up into small pieces by BPE / similar methods which have a vocabulary defined on general text. The approach includes a way to initialise the parameters of these new tokens and then fine-tuning with a focus on the layers closest to the embeddings.

Overall, reviewers were positive about the contributions. One concern was that the models used are somewhat old and recent developments may have changed the results. This is partially addressed by the new results given during the discussion period in response to reviewer questions.

Many questions raised by the reviewers were resolved in the discussion period and the changes should be incorporated into the paper. In particular:

- Adding in the new results with more recently released LLMs, (ideally including some larger scale LLMs if possible, though I appreciate the computational challenges of doing so (perhaps a library like unsloth could help?).
- Justifying the choice of baseline methods.
- Modifying some language to resolve confusion (e.g., end-to-end).
- Expanding related work to include papers discussed with reviewers.